# Enhancing Uncertainty Estimation and Interpretability via Bayesian Non-negative Decision Layer

**Xinyue Hu**[*, 1]**, Zhibin Duan**[*, 2]**, Bo Chen**[†, 1]**, Mingyuan Zhou**[3]

[1] National Key Laboratory of Radar Signal Processing, Xidian University, Xi'an, 710071, China.
[2] School of Mathematics and Statistics, Xi'an Jiaotong University, Xi'an, Shaanxi, China.
[3] McCombs School of Business, The University of Texas at Austin, Austin, TX 78712
`xinyuehu122@gmail.com` `zbduan@xjtu.edu.cn`
`bchen@mail.xidian.edu.cn`
`mingyuan.zhou@mccombs.utexas.edu`

## Abstract

Although deep neural networks have demonstrated significant success due to their powerful expressiveness, most models struggle to meet practical requirements for uncertainty estimation. Concurrently, the entangled nature of deep neural networks leads to a multifaceted problem, where various localized explanation techniques reveal that multiple unrelated features influence the decisions, thereby undermining interpretability. To address these challenges, we develop a **B**ayesian **N**on-negative **D**ecision **L**ayer (**BNDL**), which reformulates deep neural networks as a conditional Bayesian non-negative factor analysis. By leveraging stochastic latent variables, the BNDL can model complex dependencies and provide robust uncertainty estimation. Moreover, the sparsity and non-negativity of the latent variables encourage the model to learn disentangled representations and decision layers, thereby improving interpretability. We also offer theoretical guarantees that BNDL can achieve effective disentangled learning. In addition, we developed a corresponding variational inference method utilizing a Weibull variational inference network to approximate the posterior distribution of the latent variables. Our experimental results demonstrate that with enhanced disentanglement capabilities, BNDL not only improves the model's accuracy but also provides reliable uncertainty estimation and improved interpretability.

## 1 Introduction

Over the last decade, deep neural networks (DNNs) have achieved significant success and have been widely applied across various research domains (LeCun et al., 2015). As these applications expand, quantifying uncertainty in predictions has gained importance, especially for AI safety (Amodei et al., 2016). A key goal of uncertainty estimation is to ensure that neural networks assign low confidence to test cases poorly represented by training data or prior knowledge (Gal & Ghahramani, 2016). One approach to this challenge is Bayesian neural networks, which treat model parameters as random variables. Although progress has been made in developing approximate inference methods for Bayesian neural networks (Li et al., 2016; Louizos & Welling, 2017; Shi et al., 2017), computational scalability continues to pose a significant obstacle. Alternatively, deterministic methods such as deep ensembles (Lakshminarayanan et al., 2017) and dropout (Gal & Ghahramani, 2016) have been proposed. However, these approaches necessitate running full DNNs multiple times during the testing phase, resulting in high computational costs and increased inference time (Fan et al., 2021).

In addition to uncertainty estimation, there is growing demand for interpretability in DNNs, aimed at helping users understand model decisions through various tools. Advanced techniques have been developed to provide localized explanations by identifying key features or regions influencing out-

---

[*]Equal contribution.
[†]Corresponding author.

puts (Olah et al., 2017; Yosinski et al., 2015). However, a key challenge in this field is that neurons within well-trained DNNs tend to be multifaceted (Nguyen et al., 2016), meaning they respond to multiple, unrelated features. This phenomenon may arise from the entangled nature of DNNs, wherein multiple features are utilized for various tasks. To address this challenge, significant efforts have been made in the literature, including the use of specialized regularizers aimed at promoting feature disentanglement (Nguyen et al., 2016), and the employment of sparse linear decision layers to select the most important features (Wong et al., 2021). While these approaches have led to notable advancements, they warrant further exploration from both practical and theoretical standpoints. Specifically, the method of employing a sparse linear decision layer may impair task performance due to the loss of essential dimensional information. Furthermore, while empirical results suggest that sparsity enhances model interpretability and mitigates the challenges posed by multifaceted neurons (Moakhar et al., 2023), these claims lack rigorous theoretical support.

While deep neural networks (DNNs) face inherent limitations, non-negative factor analysis (NFA)—notably Poisson variants (Zhou et al., 2012)—exhibits strengths in sparse concept disentanglement and uncertainty-aware modeling through stochastic latent variables (Lee & Seung, 1999). Thus, a compelling motivation exists to transfer the good properties of NFA to DNN. However, their paradigm divergence poses challenges: NFA relies on shallow linear architectures, while DNNs employ deep non-linear hierarchies. A key insight lies in decomposing DNNs into non-linear feature extractors and linear decision layers, where recent studies show the latter is sufficient for capturing uncertainty (Kristiadi et al., 2020; Dhuliawala et al., 2023b; Joo et al., 2020; Parekh et al., 2022). Further, Wong et al. (2021) demonstrates that leveraging a sparse decision layer can enhance a model's interpretability through innovative techniques. This line of inquiry motivates the use of NFA as a linear decision layer, integrated with DNNs as feature extractors, to enhance both uncertainty estimation and model's interpretability.

With the considerations above, we developed a **B**ayesian **N**on-negative **D**ecision **L**ayer (**BNDL**), designed to empower DNNs with enhanced interpretability and uncertainty estimation capabilities. Specifically, under the categorical likelihood, the label is factorized into a gamma-distributed factor score matrix (local latent variables) and a corresponding gamma-distributed factor loading matrix (global latent variables). The former represents the latent representation of the observation, while the latter captures the interaction between the latent variables and the label. Given the challenge of intractable posterior distributions for the latent variables, we introduce a deep Weibull variational neural network to effectively approximate the gamma-distributed latent variables (Zhang et al., 2018). All parameters are trained using stochastic gradient descent (SGD) within a variational inference framework. Furthermore, we provide theoretical guarantees for the model's disentanglement capabilities, which enhances its interpretability. Additionally, our complexity analysis indicates that the increase in computational effort is minimal during both the training and uncertainty testing phases. To assess the efficacy of the proposed model, we conducted evaluations on a wide range of benchmark datasets using image classification tasks. The experimental results demonstrate that the proposed approach consistently outperforms standard classification models and offers superior uncertainty estimation. The main contributions of the paper can be summarized as follows:

- We develop a flexible **B**ayesian **N**on-negative **D**ecision **L**ayer (**BNDL**) for deep neural networks, empower its interpretability and uncertainty estimation capabilities.
- The complexity analysis shows that the computational overhead introduced by BNDL is minimal compared to DNNs. Further, we take theory analysis to verify its disentangled properties.
- We assessed the effectiveness of BNDL across multiple datasets, including CIFAR-10, CIFAR-100, and ImageNet-1k. BNDL not only preserves or even enhances baseline performance but also facilitates uncertainty estimation and improves the interpretability of neurons.

## 2 RELATED WORK

### 2.1 UNCERTAINTY ESTIMATION

Existing research in supervised learning has focused on modeling conditional distributions beyond the mean, particularly for predictive uncertainty. Ensemble methods (Liu et al., 2021; Lakshmi-

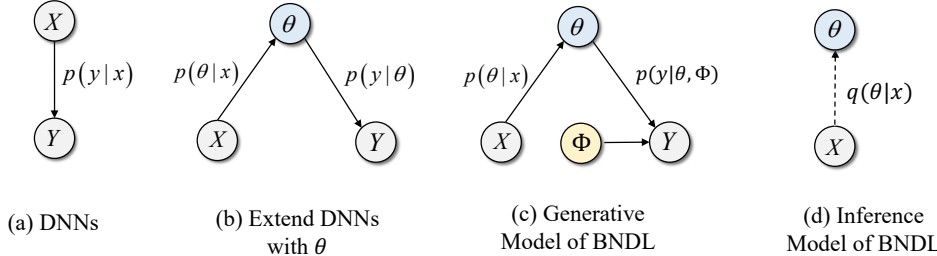

Figure 1: Illustration of the graphical models. *1(a):* the predictive process of output $Y$ for the baseline Deep Neural Network; *1(b):* the generative model of DNNs with introducing stochastic latent variable $\theta$ ; *1(c):* the generation model of the Bayesian non-negative decision layer; and *1(d):* corresponding approximate inference for latent variables $\theta$.

narayanan et al., 2017) combine neural networks with stochastic outputs to quantify uncertainty, while Bayesian neural networks (BNNs) use distributions over network parameters to reflect model plausibility (Blundell et al., 2015; Hernández-Lobato & Adams, 2015; Kingma et al., 2015; Gal & Ghahramani, 2016; Tomczak et al., 2021). However, BNNs are complex and difficult to train. In contrast, Bayesian Last Layer (BLL) methods, which focus uncertainty only on the output layer, offer a simpler, more efficient alternative (Ober & Rasmussen, 2019; Watson et al., 2021; Daxberger et al., 2021; Kristiadi et al., 2020; Harrison et al., 2024). BLL techniques, such as Ober & Rasmussen (2019)'s noise marginalization and Daxberger et al. (2021)'s use of Laplace Approximations (LA), improve probabilistic predictions. Recent advancements like retraining objectives (Weber et al., 2018) and variational improvements (Harrison et al., 2024; Watson et al., 2021) have expanded BLL's application in uncertainty estimation tasks. Another relevant approach is evidential deep learning (EDL), which uses higher-order conditional distributions, like the Dirichlet distribution, for uncertainty estimation (Sensoy et al., 2018; Malinin & Gales, 2018). EDL has shown effectiveness in tasks such as classification (Sensoy et al., 2018) and regression (Malinin et al., 2020). Furthermore, Amini et al. (2020) proposed training networks to infer hyperparameters of evidential distributions, enabling scalable uncertainty learning. Building on BLL principles, BNDL models the decision layer as a Bayesian generative model, using a gamma prior to enhance feature sparsity and disentanglement, improving both uncertainty estimation and interpretability.

## 2.2 INTERPRETABILITY TOOLS FOR DEEP NEURAL NETWORK

The goal of neural network interpretability is to identify the mechanisms underlying DNN's decision-making processes. The related research ranges from approaches which link abstract concepts to structural network components, such as specific neurons, for example via visualization (Yosinski et al., 2015; Nguyen et al., 2016), to approaches which aim to trace individual model outputs on a per-sample basis such as local surrogates (Ribeiro et al., 2016) and salience maps (Simonyan et al., 2013). However, as noted by various recent studies, these local attributions can be easy to fool or may otherwise fail to capture global aspects of model behavior (Adebayo et al., 2018; Leavitt & Morcos, 2020; Wong et al., 2021). A major confounder for interpretability is that neurons in a well-trained DNN are often multifaceted (Nguyen et al., 2016; Moakhar et al., 2023), responding to various, often unrelated, features. In contrast, our approach ensures that the identified high-level concepts—*i.e.*, the deep features utilized by the sparse decision layer—fully determine the model's behavior.

## 3 BAYESIAN NON-NEGATIVE DECISION LAYER

This section first reformulates the traditional DNNs as a latent variable model (Sec. 3.1) and then provides a detailed description of the proposed Bayesian non-negative decision layer, which consists of the generative model (Sec. 3.2) and the variational inference network (Sec. 3.3), followed by the description of the proposed variational inference algorithm (Sec. 3.4).

### 3.1 Preliminaries on deep neural networks

We first adopt a latent variable model to re-examine the DNNs, which are commonly tackled by training domain-specific neural networks with a sigmoid or softmax output layer. Dataset samples $x$ are mapped deterministically by a neural network $f$ to a real vector $\boldsymbol{z} = f(x)$, which is transformed in the softmax layer to a point on the simplex $\Delta^{|\boldsymbol{y}|}$, a discrete distribution over class labels $\boldsymbol{y} \in \mathcal{Y}$:

$$p(\boldsymbol{y} \mid x) = \frac{\exp\{\boldsymbol{z}w_{\boldsymbol{y}}^T + b_{\boldsymbol{y}}\}}{\sum_{\boldsymbol{y}' \in \mathcal{Y}} \exp\{\boldsymbol{z}w_{\boldsymbol{y}'}^T + b_{\boldsymbol{y}'}\}} \tag{1}$$

Where the $w_y$ and $b_y$ represent the weights and biases of the final fully connected layer. Thus, the classification process can be viewed as a generative process for label $y$, as shown in Fig. 1(a). Previous works (Dhuliawala et al., 2023a; Joo et al., 2020) have shown that a softmax classifier is a special case of Equation 2:

$$p(\boldsymbol{y} \mid x) = \int_{\boldsymbol{z}} p(\boldsymbol{y} \mid \boldsymbol{z})p(\boldsymbol{z} \mid x) \tag{2}$$

The neural network $f$, up to the softmax layer, models $p(\boldsymbol{z}|x)$ as a delta distribution $\delta_{\boldsymbol{z}-f(x)}$, with the softmax input representing a sample from $p(\boldsymbol{z}|x)$, and $p(\boldsymbol{y}|z)$ defined by the softmax layer. While the softmax output can be viewed as a categorical distribution, the limited randomness from the output layer is often insufficient for capturing complex dependencies (Chung et al., 2015). Additionally, DNNs with a softmax layer face overconfidence issues (Guo et al., 2017; Kristiadi et al., 2020; Liu et al., 2020), complicating uncertainty estimation. Furthermore, the data feature $\boldsymbol{z}$ and parameter $w_{\boldsymbol{y}}$ are often entangled as dense vectors, leading to multifaceted phenomena in localized explanations (Nguyen et al., 2016). These challenges motivate the development of a Bayesian non-negative decision layer.

### 3.2 Bayesian Non-negative Decomposition Layer

Building on the latent variable model of the softmax classification task, we can further reformulate DNNs as a Non-negative Factor Analysis, referred to as the **Bayesian Non-negative Decision Layer (BNDL)**. Firstly, to better capture complex dependence and aleatoric uncertainty, referring to the inherent randomness in the data that cannot be explained away, we can intuitively extend the generative model of original DNNs with stochastic latent variables $\boldsymbol{\theta}$ by modeling latent representation $z$ with a distribution. Thus, as illustrated in Fig. 1(b), the Eq. 2 is improved to

$$p(\boldsymbol{y} \mid \boldsymbol{x}) = \int_{\boldsymbol{\theta}} p(\boldsymbol{y} \mid \boldsymbol{\theta})p(\boldsymbol{\theta} \mid \boldsymbol{x}) \tag{3}$$

To further account for epistemic uncertainty, which refers to the uncertainty inherent in the model itself, we treat the weights of the final fully connected layer as stochastic latent variables. The generative model is then defined as follows, and its graphical model is shown in Fig. 1(c):

$$p(\boldsymbol{y} \mid \boldsymbol{x}) = \int_{\boldsymbol{\theta}, \boldsymbol{\Phi}} p(\boldsymbol{y} \mid \boldsymbol{\theta}, \boldsymbol{\Phi})p(\boldsymbol{\theta} \mid \boldsymbol{x})p(\boldsymbol{\Phi}) \tag{4}$$

This formulation bears similarities to factor analysis (Yu et al., 2008), where both $\boldsymbol{\theta}$ and $\boldsymbol{\Phi}$ in classical DNNs are commonly sampled from a Gaussian distribution, making the generative model akin to Gaussian factor analysis. However, while Gaussian factor analysis can effectively evaluate uncertainty, it struggles to achieve disentangled representation learning with dense latent variables (Moran et al., 2021), which is crucial for model interpretability (Nguyen et al., 2016).

Considering the gamma distribution possesses both non-linearity and non-negativity, we use the gamma distribution as the prior distribution for $\boldsymbol{\theta}$ and $\boldsymbol{\Phi}$. This choice allows for a more comprehensive capture of complex relationships within the model (Zhou et al., 2015; Duan et al., 2021; 2024), while also accommodating the characteristics of sparse non-negative variables, thereby enhancing the disentanglement and interpretability of the learned representations $\boldsymbol{\theta}$ and corresponding decision layers $\boldsymbol{\Phi}$ (Lee & Seung, 1999). Specifically, given data samples $\boldsymbol{x}_j$ and their corresponding one-hot label $\boldsymbol{y}_j \in \mathbb{R}_+^C$, where $C$ is the number of classes, we can factorize $\boldsymbol{y}_j$ under the category likelihood as follows:

$$\boldsymbol{y}_j \mid \boldsymbol{\theta}_j \sim \text{Category}\left(\boldsymbol{\theta}_j \boldsymbol{\Phi}\right), \quad \boldsymbol{\theta}_j \mid \boldsymbol{x}_j \sim \text{Gamma}\left(f_\theta(x_j), 1\right), \quad \boldsymbol{\Phi} \sim \text{Gamma}\left(1, 1\right). \tag{5}$$

where $\boldsymbol{\theta}_j \in \mathbb{R}_+^K$ is the factor score matrix, each column of which encodes the relative importance of each atom in a sample $\boldsymbol{x}_j$; and $\boldsymbol{\Phi} \in \mathbb{R}_+^{K \times C}$ is the factor loading matrix, each column of which is a factor encoding the relative importance of each term. Intuitively, in a classification problem, the $k$-th column of $\boldsymbol{\Phi} \in \mathbb{R}_+^{K \times C}$, denoted as $\boldsymbol{\phi}_k \in \mathbb{R}_+^C$, represents the $k$-th distribution across all classes. Although in our formulation the prior of the latent variable $\boldsymbol{\theta}_j$ is modulated by the input $\boldsymbol{x}_j$, the constraint can be easily relaxed to allow the latent variable to be statistically independent of the input variable (Sohn et al., 2015; Kingma et al., 2014). Therefore, we can simply define the data-independent prior of the latent variable $\boldsymbol{\theta}$ as $f_\theta(x_j) = 1$.

## 3.3 VARIATIONAL INFERENCE NEURAL NETWORK

We have constructed the generative process of $\boldsymbol{y}$, which includes the parameters $\boldsymbol{\theta}$ and $\boldsymbol{\Phi}$, in Sec. 3.2. Due to the intractable posterior in BNDL, we build a Weibull variational inference to approximate the posterior of $\boldsymbol{\theta}$ and $\boldsymbol{\Phi}$.

**Weibull Approximate Posterior:** While the gamma distribution appears suitable for the posterior distribution due to its encouragement of sparsity and adherence to the nonnegative condition, directly reparameterizing the gamma distribution can result in high noise (Zhang et al., 2018; Kingma & Welling, 2014; Knowles, 2015; Ruiz et al., 2016; Naesseth et al., 2017), and using the REINFORCE method for gradient estimation may lead to large variance (Williams, 1992). Hence, we use the reparameterizable Weibull distribution (Zhang et al., 2018) to approximate the posterior for the gamma latent variables, mainly due to the following considerations: $\boldsymbol{i}$), the Weibull distribution has a simple reparameterization so that it is easier to optimize; $\boldsymbol{ii}$) the Weibull distribution is similar to a gamma distribution, capable of modeling sparse, skewed and positive distributions. Specifically, the latent variable $x \sim \text{Weibull}(k, \lambda)$ can be easily reparameterized as:

$$x = \lambda(-\ln(1 - \varepsilon))^{1/k}, \qquad \varepsilon \sim \text{Uniform}(0, 1). \tag{6}$$

Where $\lambda$ and $k$ are the scale and shape parameter of Weibull distribution respectively; $\boldsymbol{iii}$), The KL divergence from the gamma to Weibull distributions has an analytic expression as:

$$\begin{aligned} \text{KL}\left(\text{Weibull}(k, \lambda) \| \text{Gamma}(\alpha, \beta)\right) = {} & \frac{\gamma \alpha}{k} - \alpha \log \lambda + \log k + \beta \lambda \Gamma\left(1 + \frac{1}{k}\right) \\ & - \gamma - 1 - \alpha \log \beta + \log \Gamma(\alpha) \end{aligned} \tag{7}$$

where $\gamma$ is the Euler-Mascheroni constant.

**Local latent variables Inference Network:** As shown in Fig. 1(d), the variational inference network construct the variational posterior:

$$q\left(\boldsymbol{\theta}_j \mid \boldsymbol{x}_j\right) = \text{Weibull}\left(\boldsymbol{k}_j, \boldsymbol{\lambda}_j\right) \tag{8}$$

where the inference network can be defined as :

$$\boldsymbol{k}_j = \text{Softplus}\left(\boldsymbol{f}_k(\boldsymbol{h}_j)\right), \quad \boldsymbol{\lambda}_j = \text{ReLu}\left(\boldsymbol{f}_\lambda(\boldsymbol{h}_j)\right) / \exp\left(1 + 1/\boldsymbol{k}_j\right), \quad \boldsymbol{h}_j = \boldsymbol{f}_{NN}(\boldsymbol{x}_j) \tag{9}$$

where $\boldsymbol{h}_j$ is an extracted feature with deep neural networks, such as ResNet, which can be seen as a deep feature extractor; Let $\boldsymbol{f}.(\cdot)$ denotes the neural network, where $\boldsymbol{f}_{NN}$ is the feature extractor, encompassing all layers from the input layer to the penultimate layer, and $\boldsymbol{f}_\lambda$ and $\boldsymbol{f}_k$ are the network to infer the scale and shape parameters of Weibull distribution, respectively. The Softplus function, defined as $\log(1 + \exp(\cdot))$, is applied element-wise non-linearity to ensure positive Weibull shape parameters. The Weibull distribution is used to approximate the gamma-distributed conditional posterior, and its parameters $\boldsymbol{k}_j^{(l)} \in \mathrm{R}_+^{K_l}$ and $\boldsymbol{\lambda}_j^{(l)} \in \mathrm{R}_+^{K_l}$ are inferred by the bottom-up data information using the neural networks.

**Global latent variables inference Network:** For the same reason, we also use Weibull distributions to approximate the posteriors of global latent variables $\boldsymbol{\Phi} \in \mathbb{R}_+^{K \times C}$, formulated as

$$q(\boldsymbol{\Phi} \mid -) = \text{Weibull}\left(\boldsymbol{k}_\Phi, \boldsymbol{\lambda}_\Phi\right) \tag{10}$$

where the inference network can be expressed as:

$$\boldsymbol{k}_\Phi^{(l)} = \text{Softplus}(\boldsymbol{W}_1), \quad \boldsymbol{\lambda}_\Phi^{(l)} = \text{Relu}(\boldsymbol{W}_2)/\exp\left(1 + 1/\boldsymbol{k}_\Phi\right). \tag{11}$$

Note that $\boldsymbol{W}_1$ and $\boldsymbol{W}_2$ are randomly initialized matrices with dimensions matching $\boldsymbol{\Phi}$.

**Connection with Non-Negative Matrix Factorization:** Equations 8 and 11 ensure that $\mathbb{E}[\boldsymbol{\theta}_j] = \boldsymbol{\lambda}_j$ and $\mathbb{E}[\boldsymbol{\Phi}] = \boldsymbol{\lambda}_\Phi$. Therefore, instead of sampling $\boldsymbol{\theta}_j$ and $\boldsymbol{\Phi}$ from their respective distributions, substituting their expectations makes the mapping equivalent to that of standard non-negative matrix factorization, resulting in $y_j = \boldsymbol{\lambda}_j \boldsymbol{\lambda}_\Phi$. In other words, if we let the shape parameter $k$ of the Weibull distribution approach infinity—implying that the variance of the latent variables approaches zero, and the distribution collapses into a point mass concentrated at the expectation, then the proposed stochastic decision layer reduces to non-negative matrix factorization. For further discussions on NMF, please refer to Appendix A.3.

### 3.4 Variational Inference

For BNDL, given the model parameters referred to as $\boldsymbol{\Omega}$, which consist of the parameters in the generative model and inference network, the marginal likelihood of the dataset $(X, Y)$ is defined as:

$$\boldsymbol{p}\left(Y \mid X\right) = \int \int \prod_{j=1}^{J} \boldsymbol{p}\left(\boldsymbol{y}_j \mid \boldsymbol{\theta}_j, \boldsymbol{\Phi}\right) \boldsymbol{p}\left(\boldsymbol{\theta}_j\right) d\boldsymbol{\theta}_{j=1}^{J} d\boldsymbol{\Phi} \tag{12}$$

The inference task is to learn the parameters of the generative model and the inference network. Similar to VAEs, the optimization objective of the BNDL can be achieved by maximizing the evidence lower bound (ELBO) of the log-likelihood as:

$$\begin{aligned} \mathcal{L}(Y) = &\sum_{j=1}^{J} \mathbb{E}_{q(\boldsymbol{\theta}_j \mid \boldsymbol{x}_j)} \left[\ln p\left(\boldsymbol{y}_j \mid \boldsymbol{\theta}_j, \boldsymbol{\Phi}\right)\right] \\ &- \sum_{j=1}^{J} \mathbb{E}_{q(\boldsymbol{\theta}_j \mid \boldsymbol{x}_j)} \left[\ln \frac{q\left(\boldsymbol{\theta}_j \mid \boldsymbol{x}_j\right)}{p\left(\boldsymbol{\theta}_j\right)}\right] - \mathbb{E}_{q(\boldsymbol{\Phi} \mid -)} \left[\ln \frac{q\left(\boldsymbol{\Phi} \mid -\right)}{p\left(\boldsymbol{\Phi}\right)}\right] \end{aligned} \tag{13}$$

where the first term is the expected log-likelihood of the generative model, which ensures reconstruction performance, and the last two term is the Kullback–Leibler (KL) divergence that constrains the variational distribution $q(-)$ to be close to its prior $p(-)$. The parameters in the Generalized GBN can be directly optimized by advanced gradient algorithms, like SGD (Kingma & Ba, 2015).

**Complexity analysis** Modifying the last layer of base deep neural networks minimally increases parameter count, resulting in negligible space complexity. The time complexity of ResNet is dominated by its convolutional layers and can be expressed as: $O\left(\sum_{l=1}^{L} C_{in}^{(l)} \times C_{out}^{(l)} \times H^{(l)} \times W^{(l)} \times K^{(l)} \times K^{(l)}\right)$, where $H$, $W$, $K \times K$, $C_i n$ and $C_o ut$ are the input height, width, kernel size, input channels, and output channels, respectively. In BNDL, the added time complexity from KL divergence computations for local and global latent variables is $O(C_{out}^{(L)})$ and $O(C_{out}^{(L)} \times C)$, respectively, which are negligible compared to ResNet's overall complexity. Unlike traditional ensemble methods and dropout approaches, which require multiple full-network runs to assess uncertainty, BNDL performs a single forward pass to infer the variational posterior, followed by sampling for uncertainty estimation, greatly reducing computational costs.

## 4 Theoretical Guarantees for BNDL

We provide theoretical guarantees for BNDL from the perspective of identifiable features. As described in Sec. 3.3, BNDL can be viewed as a Non-negative Matrix Factorization (NMF) problem. From this perspective, its objective function $\boldsymbol{p}\left(Y \mid \{\boldsymbol{\Phi}, \boldsymbol{\theta}, X\}\right)$ can be further reformulated as:

$$\min_{\boldsymbol{\theta} \geq 0, \boldsymbol{\Phi} \geq 0} \|Y - \boldsymbol{\theta}\boldsymbol{\Phi}\|_F^2 \tag{14}$$

In the realm of non-negative matrix factorization, many studies have aimed to establish the identifiability and uniqueness of a decomposition $\boldsymbol{\theta}\boldsymbol{\Phi}$, up to permutation and scaling. Recently, Gillis & Rajkó (2023) demonstrated that a subset of the columns of $\boldsymbol{\theta}$ and $\boldsymbol{\Phi}$ can be identified and made unique under more relaxed conditions. We will show how BNDL adheres to the criteria set forth in Gillis & Rajkó (2023), thereby enabling the learning of partially identifiable features.

**Proposition 1** ((Gillis & Rajkó, 2023)). *The $k$-th column of $\boldsymbol{\theta}$ is identifiable under the two assumptions:*

- *Selective Window: There exists a row of $\boldsymbol{\Phi}$, say the $j$-th, such that $\boldsymbol{\Phi}(j,:) = \alpha e_{(k)}^T$ for $\alpha > 0$, where $e_{(k)}^T$ represents the $k$-th standard row vector in vector space.*

- *Sparsity Constrain: The $k$-th column of $\boldsymbol{\Phi}$ contains at least $r - 1$ entries equal to zero, where $r$ is the rank of $Y$.*

The selective window assumption states that the column in $\boldsymbol{\theta}$ corresponding to $\boldsymbol{\Phi}(j,:)$ is unique in the dataset, which is reasonable in many applications (Gillis, 2020), *e.g.*, $\boldsymbol{\theta}$ can represent latent classes in a classification task, where Wang et al. (2024) achieves full identifiability by assuming each latent class corresponds to a unique sample. Under this assumption, it suffices to have a single latent class with a unique sample, making it more feasible and easier to satisfy. For the sparsity constraint, the use of a gamma prior and a ReLU activation function in $\boldsymbol{\Phi}$ within the BNDL framework, as outlined in 5 and 11, enforces sparsity during the training process. Moreover, instead of relying on the parameter norm (Wong et al., 2021), which often leads to performance degradation, we propose a more scalable and effective adaptive activation function to achieve sparsity. Specifically, we employ $f(x) = \text{ReLU}(x - \alpha)$, where $\alpha$ is a predefined constant, set as a hyperparameter. This approach offers a more flexible mechanism for inducing sparsity without sacrificing model performance. Similarly, we can demonstrate the partial identifiability of $\boldsymbol{\Phi}$, as it is often considered to be the transpose of $\boldsymbol{\theta}$ (Fu et al., 2017; HaoChen et al., 2021). In conclusion, BNDL follows the aforementioned assumptions, and its optimization objective promotes the partial identifiability of the learned features and decision layer, thereby enhancing their disentanglement capability. We validated our theoretical guarantees through experiments presented in Sec. 5.2. More details for the theoretical guarantees can be found in Appendix A.2.

## 5 EXPERIMENTS

**Experiment Setup**  Following Wong et al. (2021), we analyze the following models: (a) ResNet classifiers—ResNet-50 trained on ImageNet-1k (Deng et al., 2009; Russakovsky et al., 2015) and Places-10 (a subset of Places365 (Zhou et al., 2017)), and ResNet-18 trained on CIFAR-10/100 (Krizhevsky, 2009); (b) a ViT-based model (Dosovitskiy et al., 2021) with pretrained weights from Hugging Face*. Baselines are detailed in Sec. 5.1. We evaluate accuracy and uncertainty for quantitative analysis and use LIME for qualitative insights into BNDL predictions. Setup details, including hyperparameters and datasets, are provided in Appendix A.1. Our code is available at https://github.com/XYHu122/BNDL.

**Uncertainty evaluation metric.**  We estimate uncertainty using a hypothesis testing approach (Fan et al., 2021). This method provides interpretable $p$-values, enabling practical deployment for binary uncertainty decisions. A prediction's certainty is determined by comparing its $p$-value against a threshold. To evaluate uncertainty estimates, we use the Patch Accuracy vs. Patch Uncertainty (PAvPU) metric (Mukhoti & Gal, 2018), which defined as PAvPU = $(n_{ac} + n_{iu})/(n_{ac} + n_{au} + n_{ic} + n_{iu})$, where $n_{ac}$, $n_{au}$, $n_{ic}$, and $n_{iu}$ represent the counts of accurate-certain, accurate-uncertain, inaccurate-certain, and inaccurate-uncertain samples, respectively. Higher PAvPU values indicate that the model reliably produces accurate predictions with high certainty and inaccurate ones with high uncertainty.

**Sparsity Measurement**  We measure the sparsity of the final decision layer weights. Since the weights in our method are non-negative, we follow the approach of (Wong et al., 2021) and (Wang et al., 2024), considering weights greater than $1 \times 10^{-5}$ as non-sparse (denoted as $l_{non}$) and the remaining values as sparse (denoted as $l_{sparse}$). The sparsity is calculated as $nnz = l_{non}/(l_{non} + l_{sparse})$, where a smaller value indicates a higher level of sparsity.

### 5.1 CLASSIFICATION PERFORMANCE

Specifically, for each dataset, we conducted the following experiments:

---

*https://huggingface.co/google/vit-base-patch16-224

Table 1: Overall model accuracy across different datasets, with BNDL being our method. We use ResNet-50 as the baseline for ImageNet-1k, and ResNet-18 as the baseline for CIFAR-10 and CIFAR-100. Vit refers to vit-base-patch16-224.

| Model | CIFAR-10 | | CIFAR-100 | | ImageNet-1k | |
|---|---|---|---|---|---|---|
| | ACC | PAvPU | ACC | PAvPU | ACC | PAvPU |
| ResNet | $94.98 \pm 0.12$ | - | $74.62 \pm 0.23$ | - | $75.33 \pm 0.14$ | - |
| MC Dropout | $94.54 \pm 0.03$ | $78.83 \pm 0.12$ | $78.12 \pm 0.06$ | $64.41 \pm 0.22$ | $75.98 \pm 0.08$ | $76.50 \pm 0.02$ |
| BM | $94.07 \pm 0.07$ | $93.98 \pm 0.3$ | $75.81 \pm 0.34$ | $77.13 \pm 0.67$ | - | - |
| CARD | $90.93 \pm 0.02$ | $91.11 \pm 0.04$ | $71.42 \pm 0.01$ | $71.48 \pm 0.03$ | $76.20 \pm 0.00$ | $76.29 \pm 0.01$ |
| ResNet-BNDL | $\mathbf{95.54 \pm 0.08}$ | $\mathbf{95.58 \pm 0.20}$ | $\mathbf{79.82 \pm 0.13}$ | $\mathbf{81.1 \pm 0.21}$ | $\mathbf{77.01 \pm 0.14}$ | $\mathbf{77.66 \pm 0.03}$ |
| ViT-Base | $95.51 \pm 0.03$ | - | $84.15 \pm 0.03$ | - | $80.33$ | - |
| ViT-BNDL | $\mathbf{96.34 \pm 0.04}$ | $\mathbf{97.01 \pm 0.02}$ | $\mathbf{85.16 \pm 0.03}$ | $\mathbf{86.37 \pm 0.11}$ | $\mathbf{81.29 \pm 0.02}$ | $\mathbf{82.50 \pm 0.03}$ |

*(a) Performance and Uncertainty Evaluation* We replaced the decision layers in the network with BNDL and performed supervised training from scratch. The results are shown in Table 1. The baseline models are grouped into two categories: *1)* Uncertainty estimation networks, including Bernoulli MC Dropout(Gal & Ghahramani, 2016), BM (Joo et al., 2020) and CARD (Han et al., 2022) *2)* Dense decision layer baselines: including ViT-Base (Dosovitskiy et al., 2021) (We used the pretrained weights for the vit-base-patch16-224 only, modifying the decision layer for continued training.) and ResNet (He et al., 2016). It is important to note that the goal of BNDL is not to achieve a substantial improvement in the model's performance, but rather to preserve the model's performance while enhancing its interpretability and uncertainty estimation capabilities.

*(b) Impact of Sparsity.* We replaced the decision layers of a pre-trained model with BNDL, froze the existing feature layers, and fine-tuned only the parameters of the BNDL. The results across different datasets are illustrated in Fig. 3, where we compare BNDL (shown in blue) with the Debuggable Network (Wong et al., 2021) (shown in orange), both of them utilize the same backbone with different sparse decision layers.

### 5.1.1 PERFORMANCE AND UNCERTAINTY EVALUATION

In Table 1 , we show accuracy and PAvPU. Our model reports the mean and variance across 5 different random seeds, while the results of other models are reported from previous papers if available. Since we directly used the source* to test on ImageNet-1k, the variance term is not provided in the table. We can observe that: *1)* By leveraging stochastic latent variables to capture complex dependencies, BNDL consistently outperformed all datasets and demonstrated improved performance across various widely-used architectures, including ResNet and ViT. *2)* The integration of BNDL endowed the model with the capability for uncertainty estimation, as evidenced by the improvements in PAvPU metrics when compared to several strong baselines. *3)* BNDL exhibits scalability and can be extended to larger datasets, such as ImageNet-1k, as demonstrated by the complexity analysis.

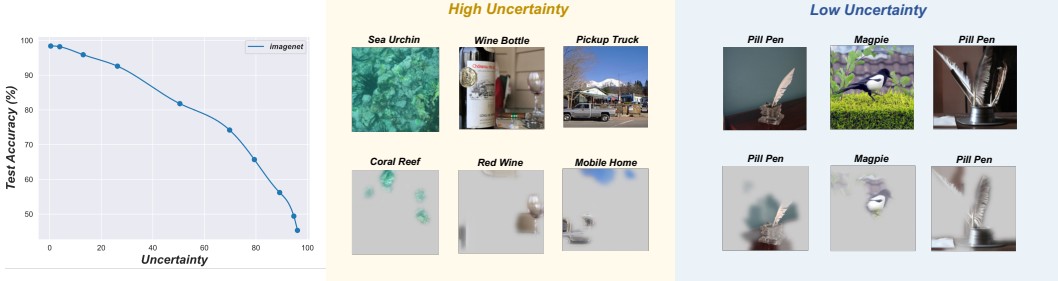

Figure 2: The leftmost line chart illustrates the average uncertainty and accuracy across subsets of the ImageNet test set. The middle and right panels sample images from the subsets with the highest and lowest uncertainty, as defined by the curve. The top row shows the original images with ground truth labels, while the bottom row displays the model's predictions alongside LIME visualizations.

**Relation between Uncertainty and Accuracy**   We conducted an ablation study to explore the relationship between prediction uncertainty and downstream performance. Using ResNet-BNDL, we sorted the ImageNet test set by evaluation uncertainty into 10 subsets. For each subset, we calculated the average accuracy and uncertainty, then plotted the results in Fig. 2. We also selected images from the highest and lowest uncertainty subsets and visualized their LIME explanations, showing the most influential features of the activations on the right side of the figure. The line chart illustrates a clear ***negative correlation*** between uncertainty and accuracy: higher uncertainty corresponds to lower accuracy. This suggests that the model provides reliable uncertainty estimates, helping to avoid potential misclassifications. In the visualization, we observe that the model made correct predictions for images with low uncertainty, while for images with high uncertainty, the visualizations reveal the causes of misclassification, *e.g.*, in the image of a wine bottle, the model primarily focused on the wine glass filled with red wine in the background, leading to a misclassification as red wine. The Uncertainty Vs Acc results for additional datasets and ViT-BNDL are provided in the Appendix A.3.

### 5.1.2   IMPACT OF SPARSITY

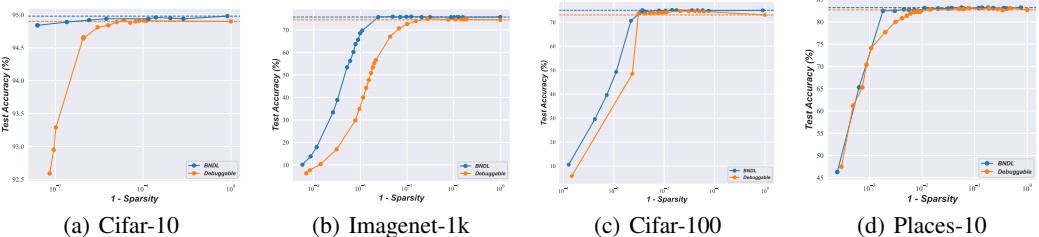

|          (a) Cifar-10          |          (b) Imagenet-1k          |          (c) Cifar-100          |          (d) Places-10          |

Figure 3: Sparsity-accuracy trade-offs for BNDL and Debuggable Network (Wong et al., 2021). Each point on the curve represents a BNDL classifier. Horizontal dashed lines indicate the fully dense accuracy for each network. The x-axis shows the proportion of non-sparse weights, with higher values indicating denser distributions, while the y-axis represents test accuracy on each dataset.

The results of fine-tuning on the original model are shown in Fig. 3. For each dataset, we control the sparsity according to the activation function mentioned in Eq. 11, resulting in the Sparsity vs. accuracy curve. The blue lines represent our method, while the orange lines represent the Debuggable Network (Wong et al., 2021), which also employs the idea of a sparse decision layer. It can be observed that at the same level of sparsity, our model overall outperforms the Debuggable Network across different datasets. Additionally, across different datasets, the decision layer can be made substantially sparser—by up to two orders of magnitude—with minimal impact on accuracy (cf. Fig. 3(a)). For instance, when the sparsity of the ImageNet-1k classifier is 0.0024, the network's classification accuracy still reaches 75.7%.

### 5.2   INTERPRETABILITY EVALUATION

**Disentangled representation learning**   To validate the disentanglement on real-world data, we adopt an unsupervised disentanglement metric SEPIN@$k$ (Do & Tran, 2019). SEPIN@$k$ measures how each feature $\boldsymbol{\theta}_i$ is disentangled from others $\boldsymbol{\theta}_{\neq i}$ by computing their conditional mutual information with the top $k$ features, *i.e.*, SEPIN@$k = 1/k \sum_{i=1}^{k} I(x, \boldsymbol{\theta}_{r_i}|\boldsymbol{\theta}_{\neq r_i})$, which are estimated with InfoNCE lower bound (Oord et al., 2018) implemented following (Wang et al., 2024).

Table 2: Feature disentanglement score on Imagenet-1k, where @k denotes the top-k dimensions. Values are scaled by $10^2$, we use ResNet-50 as the baseline for BNDL

|              | SEPIN@1 | SEPIN@10 | SEPIN@100 | SEPIN@1000 | SEPIN@all |
|--------------|---------|----------|-----------|------------|-----------|
| ResNet50     | $1.50 \pm 0.02$ | $1.03 \pm 0.01$ | $0.60 \pm 0.01$ | $0.31 \pm 0.01$ | $0.23 \pm 0.01$ |
| ResNet50-BNDL | **$2.59 \pm 0.03$** | **$2.12 \pm 0.01$** | **$1.30 \pm 0.01$** | **$0.65 \pm 0.01$** | **$0.44 \pm 0.01$** |

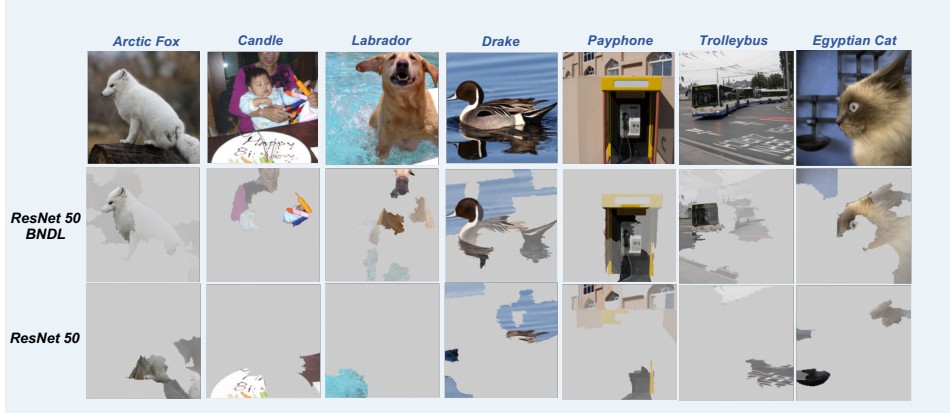

Figure 4: The LIME visualizations for BNDL and ResNet-50, focusing on the largest $\theta$ for each image, show that BNDL's features align more closely with the semantic meaning of true labels, suggesting more disentangled representations. As we selected the top-10 super-pixels for visualization, the results may include some less significant super-pixels; this issue is alleviated when we reduce the number of top-$k$ super-pixels.

As shown in Table 2, BNDL features exhibit much better disentanglement than ResNet-50 across all top-$k$ dimensions. The advantage is more pronounced when considering the top features, as learned features also contain noise dimensions. This verifies the disentanglement of learned features, as analyzed in Sec. 4. BNDL indeed provides better feature disentanglement on real-world data. The disentanglement results on other datasets can be found in Appendix A.3.

**Visualization** We visualized the feature representation $\theta$ of BNDL and the baseline model (ResNet-50) for the same images in ImageNet-1k, as illustrated in Fig. 4. Specifically, we selected the feature $\theta$ with the highest activation for each image and applied the LIME visualization, in line with the approach used in Fig. 2. The top row of Fig. 4 shows the true categories of the corresponding images, the second row presents our visualization results, and the third row displays the visualization results of the baseline model. Overall, the visualization results of BNDL are more semantically meaningful compared to those of ResNet-50. For instance, in the image of a candle, BNDL successfully captures parts of the candle, while ResNet-50 only identifies the cake. Similar observations occur in other categories, and we provide additional visualization results in Appendix A.3. This finding aligns with the conclusions drawn in 4, suggesting that BNDL has learned more identifiable features through the constraint of sparsity.

## 6 CONCLUSION

We introduce BNDL as a simple and scalable Bayesian decision layer that excels in both uncertainty estimation and interpretability, while maintaining or improving accuracy across a range of tasks, including large-scale applications. With an efficient parameterization of the covariance-dependent variational distribution, BNDL enhances the flexibility of DNNs with only a slight increase in memory and computational cost. We demonstrate the broad applicability of BNDL on both ResNet-based and ViT-based models and show that BNDL achieves superior performance compared to these baselines. Notably, we provide both practical and theoretical guarantees for BNDL's ability to learn more disentangled and identifiable features. Based on these results, we believe BNDL can serve as an efficient alternative to decision layer in the versatile tool box of modules.

## REPRODUCILTLY STATEMENT

The novel methods introduced in this paper are accompanied by detailed descriptions (Sec. 3), and their implementations are provided at https://github.com/XYHu122/BNDL.

## ACKNOWLEDGMENTS

The work of X. Hu, Z. Duan, and B. Chen was supported in part by the National Natural Science Foundation of China under Grant U21B2006; in part by Shaanxi Youth Innovation Team Project; in part by the Fundamental Research Funds for the Central Universities QTZX23037 and QTZX22160; and in part by the 111 Project under Grant B18039.

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

# A    APPENDIX / SUPPLEMENTAL MATERIAL

## A.1    EXPERIMENTAL SETTING

All experiments are conducted on Linux servers equipped with 32 AMD EPYC 7302 16-Core Processors and 2 NVIDIA 3090 GPUs. Models are implemented in PyTorch version 1.12.1, scikit-learn version 1.0.2 and Python 3.7. The CIFAR-10, CIFAR-100, and ImageNet-1k datasets we used are all publicly available standard datasets. As for Places-10, it is a subset of Places365 (Zhou et al., 2017) containing the classes "airport terminal", "boat deck", "bridge", "butcher's shop", "church-outdoor", "hotel room", "laundromat", "river", "ski slope" and "volcano". We chose this dataset to ensure a fair comparison with DebuggableNetworks (Wong et al., 2021), as they also selected this subset.

- **Training from scratch setup** For ResNet-18 on CIFAR-10 and CIFAR-100, we set the batch size to 128, learning rate to 0.1, training epochs to 150, and weight decay to 5e-4. For ResNet-50 on ImageNet-1k, we set the batch size to 256, weight decay to 1e-4, epochs to 200, and learning rate to 0.1. For the calculation of PAvPU, we sampled 20 times to obtain 20 different logits, and then selected the top-2 classes based on their mean logits, performing a two-sample t-test to obtain the p-value. Following the approach used by CARD and MC Dropout models, we set the p-value threshold to 0.05 to determine whether a sample is classified as certain.

- **Fine-tuning setup** For Places-10, we set the learning rate to 0.1, batch size to 128, and epochs to 100. For ImageNet-1k, CIFAR-10, and CIFAR-100, we set the learning rate to 0.001 and epochs to 200.

- **Sparsity Vs Accuracy in Sec. 5.1.2** We follow the default settings of (Wong et al., 2021) when running the Debuggable baselines, and we run BNDL using the settings from the fine-tuning setup. It is worth mentioning that in Debuggable Networks, the sparsity of the decision layer is controlled via the elastic net, which adjusts the sparsity of decision layer weights through the regularization path, this kind of parameter norm often leads to performance degradation. In contrast, BNDL increases the sparsity of the decision layer by using a gamma distribution as the prior. Additionally, we control the sparsity of the weights by applying an activation function to the weights, $w' = \mathrm{ReLU}(w - \alpha)$, where $\alpha$ is a predefined constant set as a hyperparameter.

**Visualization tool (LIME)**    Traditionally, LIME is used to obtain instance-specific explanations—i.e., to identify the super-pixels in a given test image that are most responsible for the model's prediction. In our setting, we follow this intuition and use the following two step-procedure to obtain LIME-based feature interpretations: *(i)* We randomly select an image category and then randomly choose $K$ images from that category. For each image, we identify the feature with the maximum activation for visualization. Among them, our definition of maximum activation is that we select the maximum weight of the category index corresponding to the image in $\mathbf{\Phi}$, and select the feature neuron multiplied by this weight. *(ii)* Run LIME on each of these examples to identify relevant super-pixels. At a high level, this involves performing linear regression to map image super-pixels to the (normalized) activation of the deep feature (rather than the probability of a specific class as is typical).

## A.2    THEORETICAL GUARANTEES FOR BNDL

**Problem Formulation**    As described in Sec. 3.3, BNDL can be viewed as a Non-negative Matrix Factorization (NMF) problem. From this perspective, its objective function $p\left(Y \mid \{\mathbf{\Phi}, \boldsymbol{\theta}, X\}\right)$ can be further reformulated as:

$$\min_{\boldsymbol{\theta} \geq 0, \mathbf{\Phi} \geq 0} \|Y - \boldsymbol{\theta}\mathbf{\Phi}\|_F^2 \tag{15}$$

This property enables NMF to accurately recover the ground truth factors that generated the data. Following the definition in (Gillis & Rajkó, 2023), we first define an exact NMF (that is, an errorless reconstruction) as follows:

**Definition 1.** *(Excat NMF of size r) Given a nonnegative matrix $Y \in \mathbb{R}^{m \times n}$, the decomposition $\boldsymbol{\theta}\mathbf{\Phi}$ where $\boldsymbol{\theta} \in \mathbb{R}_+^{m \times r}$ and $\mathbf{\Phi} \in \mathbb{R}_+^{r \times n}$ is an exact NMF of $Y$ of size $r$ if $Y = \boldsymbol{\theta}\mathbf{\Phi}$.*

The formally defined identifiability of an Exact NMF is as follows:

**Definition 2.** *(Identifiability of Exact NMF) The Exact NMF of $Y = \boldsymbol{\theta}_*\boldsymbol{\Phi}_*$ of size $r$ is identifiable if and only if for any other Exact NMF of $Y = \boldsymbol{\theta}\boldsymbol{\Phi}$ of size $r$, there exists a permutation matrix $\Pi \in \{0, 1\}^{r \times r}$ and a nonsingular diagonal scaling matrix $D$ such that:*

$$\boldsymbol{\theta} = \boldsymbol{\theta}_*\Pi D \quad and \quad \boldsymbol{\Phi} = D^{-1}\Pi\boldsymbol{\Phi}_* \tag{16}$$

Intuitively, Definition 2 indicates that all columns of $\boldsymbol{\theta}$ and $\boldsymbol{\Phi}$ must be identifiable. Achieving this typically requires very stringent conditions, such as the requirement for both $\boldsymbol{\theta}$ and $\boldsymbol{\Phi}$ to satisfy the so-called sufficiently scattered condition (SSC) (Huang et al., 2013). Therefore, we concentrate on the partial identifiability of BNDL, which similarly ensures the identifiability and uniqueness of a subset of columns of $\boldsymbol{\theta}$ and $\boldsymbol{\Phi}$ under more relaxed conditions.

**Partially Identifiable Features** To demonstrate that BNDL is partially identifiable, we first present the definition of partial identifiability in exact NMF.

**Definition 3.** *(Partial identifiability in Exact NMF) Let $Y = \boldsymbol{\theta}_*\boldsymbol{\Phi}_*$ be an exact NMF of $Y$ of size $r$. The $k$-th column of $\boldsymbol{\theta}_*$ is identifiable if and only if for any other Exact NMF of $Y = \boldsymbol{\theta}\boldsymbol{\Phi}$ of size $r$, there exits an index set $j$ and a scalar $\alpha > 0$ such that:*

$$\boldsymbol{\theta}(:, j) = \alpha\boldsymbol{\theta}_*(:, k) \tag{17}$$

*Similarly, we can define the identifiability of the $k$th column of $\boldsymbol{\Phi}_*$ using symmetry.*

Previous work (Gillis & Rajkó, 2023) has shown that Definition 3 can hold under two relatively relaxed assumptions.

**Proposition 2.** *(Gillis & Rajkó, 2023) Let $Y = \boldsymbol{\theta}_*\boldsymbol{\Phi}_*$ where $\boldsymbol{\theta}_* \in \mathbb{R}_+^{m \times r}$ and $\boldsymbol{\Phi}_* \in \mathbb{R}_+^{r \times n}$ with rank$(Y) = r$. Without loss of generality, assume $Y, \boldsymbol{\Phi}_*$ and $\boldsymbol{\theta}$ are column stochastic. The $k$-th column of $\boldsymbol{\theta}_*$ is identifiable if it satisfies the following two conditions:*

- *(**Selective Window**) There exists a row of $\boldsymbol{\Phi}_*$, say the $j$-th, such that $\boldsymbol{\Phi}_*(j, :) = \alpha e_{(k)}^T$ for $\alpha > 0$.*

- *(**Sparsity Constrain**) There exists a subset $\mathcal{J}$ of $r - 1$ columns of $Y$, namely $Y(:, \mathcal{J})$, such that $rank(Y(:, \mathcal{J})) = r - 1$ and for all $j \in \mathcal{J}$:*

$$\mathcal{F}_{\boldsymbol{\theta}_*}(\boldsymbol{\theta}_*(:, k)) \cap \mathcal{F}_{\boldsymbol{\theta}_*}(R(:, j)) = \emptyset \tag{18}$$

  *which means that the minimal face containing the $k$-th column of $\boldsymbol{\theta}_*$ does not intersect with the minimal faces containing the columns of $Y(:, \mathcal{J})$.*

$i$)**For the Selective Window assumption**: Intuitively, this assumption means that the column in $\boldsymbol{\theta}$ (latent class) corresponding to the $\boldsymbol{\Phi}_*(j, :)$ appears uniquely in the dataset, which is reasonable in many applications (Gillis, 2020). E.g., in classification tasks (Wang et al., 2024), the authors achieve full identifiability by positing that each latent class has a unique sample. In the context of selective window assumption, we only need to assume the presence of a single latent class with a unique sample to satisfy the selective window assumption, which makes it more feasible and easier to achieve. $ii$)**For the Sparsity Constrain**: This condition implies that the $k$-th column of $\boldsymbol{\Phi}_*$ contains at least $r - 1$ entries equal to zero, namely, $\boldsymbol{\Phi}_*(\mathcal{J}, k) = 0$. Due to the use of a gamma prior and a ReLU function in $\boldsymbol{\Phi}$ within BNDL, as shown in 5 and 11 respectively, $\boldsymbol{\Phi}$ is enforced to be sparse during the training process. Additionally, in Sec. 5.1.1, we demonstrate the high sparsity of BNDL, for instance, the 1-sparsity of decision weights for ImageNet is only 0.04. Only a small portion of the weights have a decisive impact on the final results, indicating that BNDL satisfies the sparsity constraint.

In summary, our theory demonstrates that BNDL satisfies the assumptions outlined in 2, facilitating the partial identifiability of the learned features.

## A.3 Additional Experimental Results

**Uncertainty Evaluation** We provide additional ablation study results on Fig. 5, with experimental settings consistent with Sec. 5.1.1. For plotting the curves, we used B-spline interpolation to generate smooth curves, setting $k = 3$ (cubic spline). The line charts illustrate a clear negative correlation

between uncertainty and accuracy: higher uncertainty corresponds to lower accuracy. This suggests that the model provides reliable uncer- tainty estimates, helping to avoid potential misclassifications.

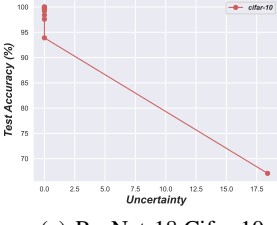

(a) ResNet-18 Cifar-10

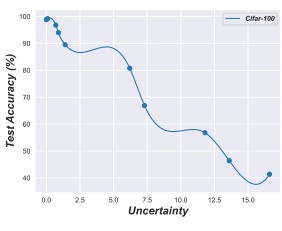

(b) ResNet-18 Cifar-100

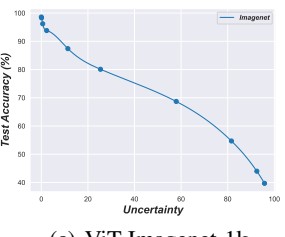

(c) ViT Imagenet-1k

Figure 5: Uncertainty Vs Test Acc Curve on other datasets and model.

Table 3: Feature score on Cifar-100, where @k denotes the top-k dimensions. Values are scaled by $10^2$

| Cifar-100 | SEPIN@1 | SEPIN@10 | SEPIN@100 | SEPIN@1000 | SEPIN@all |
|---|---|---|---|---|---|
| ResNet18 | $3.17 \pm 0.06$ | $2.40 \pm 0.03$ | $1.79 \pm 0.03$ | $1.21 \pm 0.01$ | $1.21 \pm 0.01$ |
| BM | $3.19 \pm 0.05$ | $2.53 \pm 0.02$ | $1.17 \pm 0.02$ | $1.28 \pm 0.02$ | $1.24 \pm 0.01$ |
| ResNet18-NMF | $3.21 \pm 0.03$ | $2.95 \pm 0.02$ | $1.87 \pm 0.03$ | $1.33 \pm 0.02$ | $1.23 \pm 0.03$ |
| BNDL | $\mathbf{3.91 \pm 0.06}$ | $\mathbf{3.43 \pm 0.03}$ | $\mathbf{2.77 \pm 0.03}$ | $\mathbf{1.69 \pm 0.03}$ | $\mathbf{1.69 \pm 0.02}$ |

**Disentangled Measurement** Disentanglement Result on BNDL, BM, ResNet-NMF and ResNet18. Among them, ResNet-NMF represents a framework where non-negativity constraints are applied to both the weights of the decision layer and the input features, transforming the problem into one of non-negative matrix factorization. It can be observed that BNDL consistently achieves the best disentanglement metrics, aligning with the conclusions drawn in Section 5.2 of the paper. This suggests that BNDL has successfully learned more identifiable features through the constraint of sparsity.

Table 4: Accuracy and weight sparsity results of Cifar-100, where ResNet-NMF applied non-negativity constraints to both the input features and weights of decision layer.

| Cifar-100 | Accurarcy | 1-Sparsity |
|---|---|---|
| ResNet18 | $74.62 \pm 0.23$ | $0.97 \pm 0.02$ |
| ResNet18-NMF | $76.73 \pm 0.16$ | $0.23 \pm 0.01$ |
| ResNet18-BNDL | $\mathbf{79.82 \pm 0.13}$ | $\mathbf{0.12 \pm 0.01}$ |

**Additional Discussion on the relation of BNDL and Non-negative Matrix Factorization** We now provide a more detailed explanation of the relationship between NMF and interpretability. Specifically, the non-negativity constraint in NMF ensures that the decomposition of the data matrix can be interpreted as an "additive combination" rather than a complex mathematical operation involving cancellations between positive and negative terms. This characteristic aligns more closely with how humans naturally understand patterns in real-world data. Therefore, both BNDL and Wang et al. (2024) leverage the principles of NMF by modeling the final layer of the network as an NMF problem, employing non-negativity constraints to enhance interpretability.

However, there remain key differences between BNDL and Wang et al. (2024), which include the following: 1) NCL remains a point-estimation model that enforces non-negativity constraints by applying a ReLU activation function to the features. In contrast, BNDL probabilistically models the features as non-negative distributions, which enables uncertainty estimation—something that NCL

does not accommodate; 2) Although NMF naturally provides some degree of disentanglement, it is often insufficient for handling complex data (Hoyer, 2004), necessitating additional constraints to improve this effect. Unlike NCL, which does not introduce such extra constraints, BNDL applies a gamma prior to the factors in matrix decomposition, further enhancing the sparsity and non-linearity of these factors. This additional constraint strengthens both the disentanglement and interpretability of the model.

Regarding (Duan et al., 2024), it also recognizes the additive property introduced by non-negativity, which aids in the disentanglement of the network. It leverages the non-negativity and sparsity of the gamma distribution to design a Variational Autoencoder (VAE) generative model, achieving better disentanglement performance compared to Gaussian-VAE. In general, while BNDL shares some similar tools with (Wang et al., 2024) and (Duan et al., 2024), there are notable differences in their network designs and architectures, as well as the distinct objectives they serve in different tasks.

we incorporated NMF into a supervised learning framework and conducted experiments on CIFAR-100. Consistent with the experimental settings in the paper, ResNet-NMF adopts the ResNet18 baseline model. The experiments were performed on an NVIDIA 3090 GPU. We ran five experiments with different random seeds and computed the mean and variance of the results. The experimental outcomes are presented in the Table 3 and 4. From the experiments on accuracy and disentanglement, the following observations can be made: (1) Incorporating NMF into the network enhances its performance. By introducing non-negativity constraints on both weights and features, ResNet-NMF achieves more sparse weights and improved disentanglement performance. (2) BNDL consistently achieves the best results on the CIFAR-100 dataset. This can be attributed to its probabilistic modeling and the additional sparsity and nonlinear constraints introduced by the gamma prior. These factors lead to greater weight sparsity and further improved disentanglement, aligning with our discussion.

**Additional Visualization Results**    We visualized the feature representation $\theta$ of BNDL and the baseline model (ResNet-50) for the same images in ImageNet-1k, as illustrated in Figures 8 to 12. Specifically, we selected the feature $\theta$ with the highest activation for each image and applied the LIME method using the top-10 super-pixels for visualization, in line with the approach used in Fig. 2. We have added comparative visualization results of BNDL and several models on ImageNet-1k, including the uncertainty estimation model BM (Joo et al., 2020) and the sparse decision-layer model Debuggable Networks (Wong et al., 2021), the results are shown in Fig. 6. The top row of Figures shows the true categories of the corresponding images, the second row presents our visualization results, and the third row displays the visualization results of the baseline model. Overall, the visualization results of BNDL are more semantically meaningful compared to those of ResNet-50.

We also included the results of post-hoc explanations using GradCAM(Selvaraju et al., 2017), which is based on the concept of Class Activation Mapping. It aims to generate heatmaps by exploiting the relationship between the feature maps of specific layers in the neural network and the final prediction of the classification task, visually highlighting the regions that contribute the most to the prediction of a particular class. In our experiment, we used the most activated class for each image to generate explanations, with the results shown in Fig. 7. It can be observed that BNDL tends to generate more focused heatmaps. For example, in the first image, BNDL only focuses on the region of the dog, while ResNet50 also attends to the surrounding ground. These results further demonstrate that BNDL is inclined to generate more disentangled features.

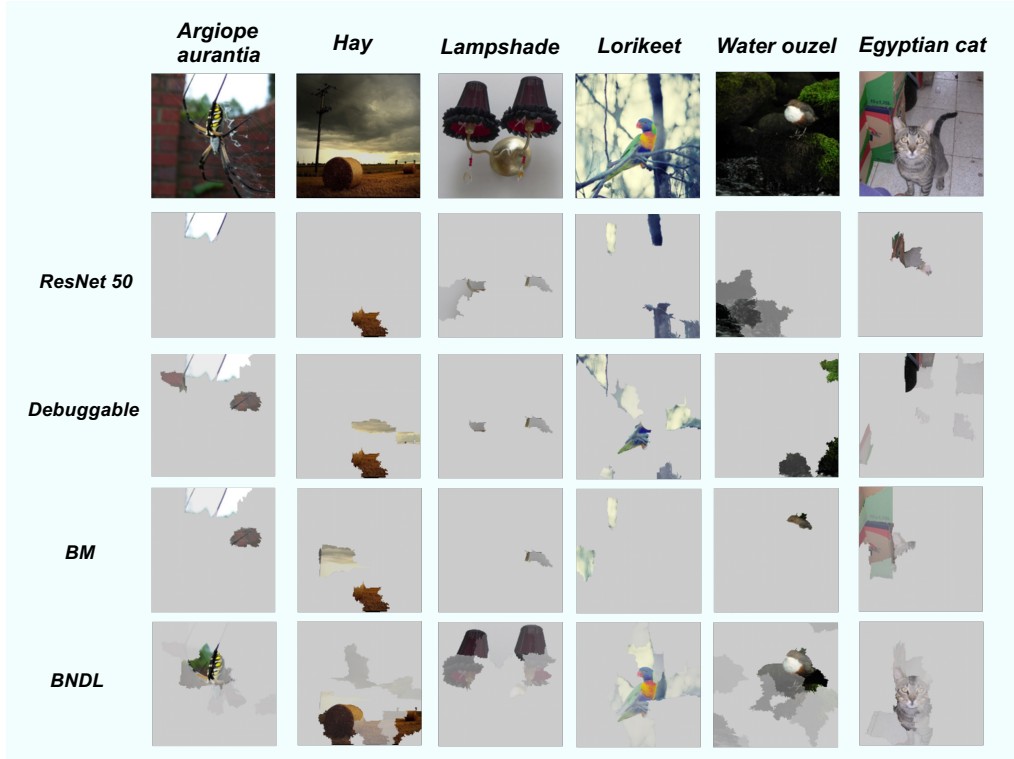

Figure 6: The LIME visualization results for BNDL ,ResNet-50 Debuggable Networks and BM, focusing on the largest $\theta$ for each image, demonstrate that BNDL's feature visualization aligns more closely with the semantic meaning of the true labels. This suggests that BNDL has learned more identifiable features.

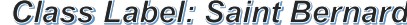

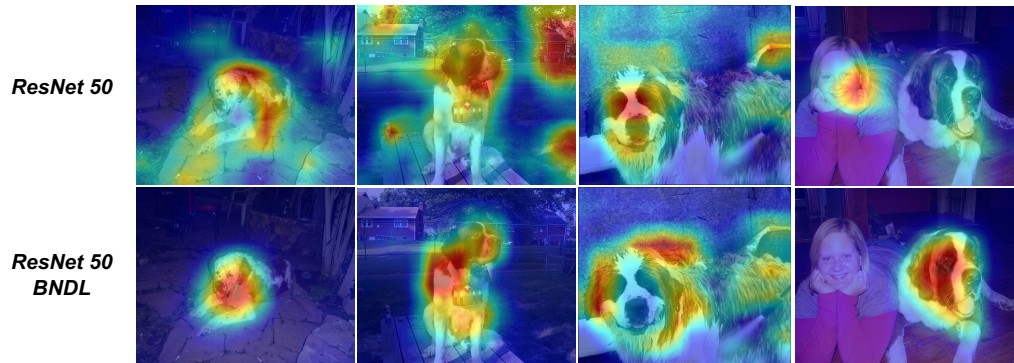

Figure 7: We applied GradCAM to visualize ResNet50 and BNDL, and the corresponding heatmaps are shown above. The ground truth label for the visualized image is "Saint Bernard." It can be observed that the BNDL visualization is more focused, while ResNet exhibits multifaceted behavior. For instance, in the fourth image, ResNet attends to both the person and the dog, whereas BNDL is more disentangled, focusing solely on the dog.

**Arctic fox, white fox, Alopex lagopus**

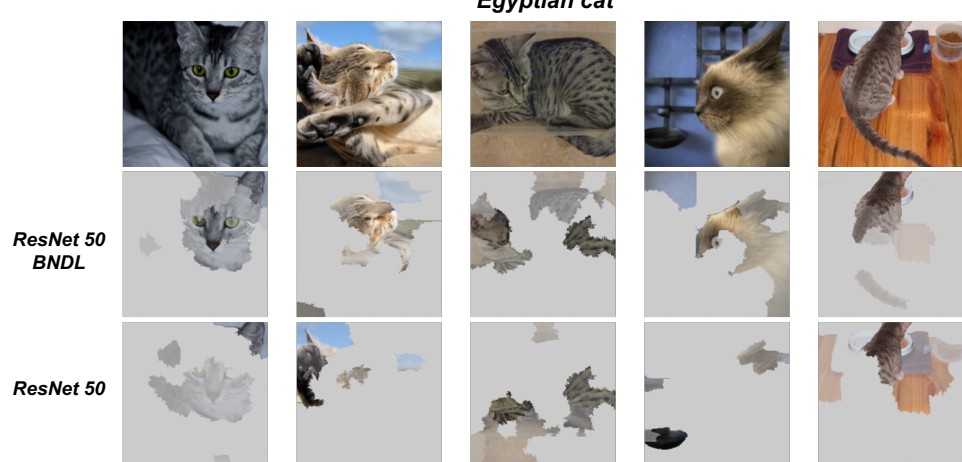

Figure 8: The LIME visualization results for BNDL and ResNet-50, focusing on the largest $\theta$ for each image, demonstrate that BNDL's feature visualization aligns more closely with the semantic meaning of the true labels compared to ResNet-50. This suggests that BNDL has learned more identifiable features.

**Egyptian cat**

Figure 9: The LIME visualization results for BNDL and ResNet-50, focusing on the largest $\theta$ for each image, demonstrate that BNDL's feature visualization aligns more closely with the semantic meaning of the true labels compared to ResNet-50. This suggests that BNDL has learned more identifiable features.

**loupe, jeweler's loupe**

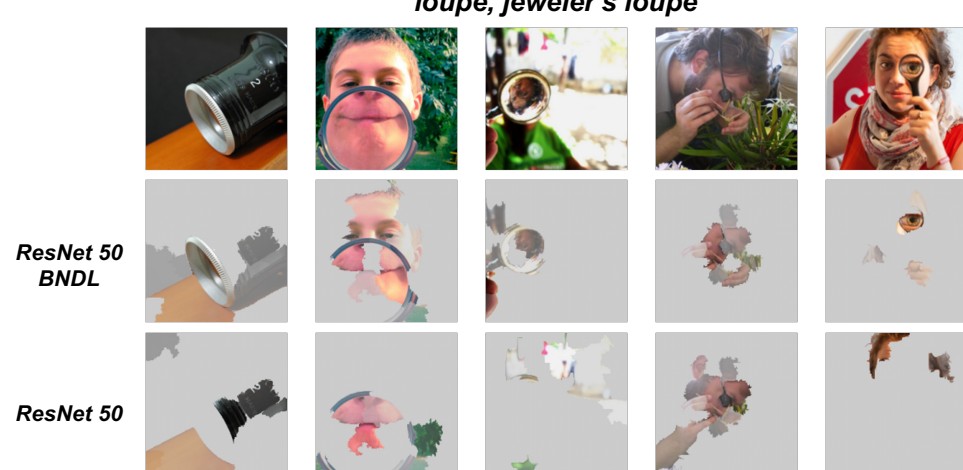

Figure 10: The LIME visualization results for BNDL and ResNet-50, focusing on the largest $\theta$ for each image, demonstrate that BNDL's feature visualization aligns more closely with the semantic meaning of the true labels compared to ResNet-50. This suggests that BNDL has learned more identifiable features.

**Redbone**

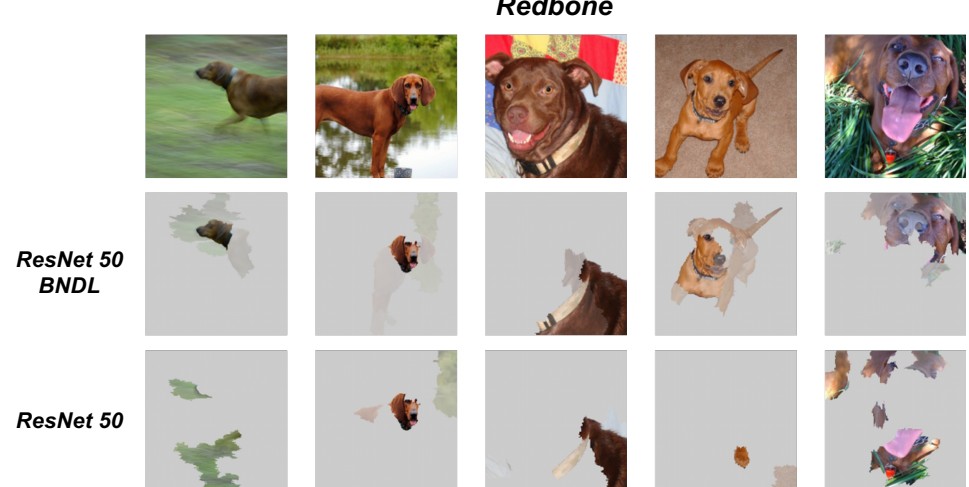

Figure 11: The LIME visualization results for BNDL and ResNet-50, focusing on the largest $\theta$ for each image, demonstrate that BNDL's feature visualization aligns more closely with the semantic meaning of the true labels compared to ResNet-50. This suggests that BNDL has learned more identifiable features.

Figure 12: The LIME visualization results for BNDL and ResNet-50, focusing on the largest $\boldsymbol{\theta}$ for each image, demonstrate that BNDL's feature visualization aligns more closely with the semantic meaning of the true labels compared to ResNet-50. This suggests that BNDL has learned more identifiable features.

