# OpenReview forum: "Enhancing Uncertainty Estimation and Interpretability with Bayesian Non-negative Decision Layer"
_ICLR.cc/2025/Conference — ICLR 2025 Poster_

### Official Review · Reviewer_dpRV · 2024-11-03

**Soundness:** 3
**Presentation:** 3
**Contribution:** 2
**Rating:** 6
**Confidence:** 3

**Summary:**

The manuscript suggests using Bayesian non-negative decision layer for improving model's uncertainty evaluation and sparsity (disentanglement), with no (statistically significant) loss of accuracy.

**Strengths:**

Suggested BDNL seems advantageous for uncertainty evaluation and disentangled representation learning.

The authors try to perform theoretical analysis of their method.

For most paragraphs ZeroGPT score was 0%, for some 4% and 8%. Thus rather human-written.

**Weaknesses:**

Literature overview: why you do not cite works on DNNs and non-negative factor analysis in the interpretability framework, these are probably the closest works to your manuscript and constitute the core of your work? E.g.:
https://proceedings.neurips.cc/paper_files/paper/2022/hash/e53280d73dd5389e820f4a6250365b0e-Abstract-Conference.html

Theorem 1 is not a theorem (please check any statistical/ML literature, like AoS or NeurIPS for what is a theorem), neither is its "proof" is a proof, this is just a discussion. I would suggest you change the presentation.

The improvement of the performance does not seem to be significant at all, which is Ok, but the PAvPU might still seem questionable. How many receptions have been performed, what are your p-values? How, e.g., these numbers in Table 1 with +/- in front were calculated?

There are typos in the manuscript (which means it is human-written).

**Questions:**

The results on both uncertainty and disentanglement use one metric only each. Would it be possible include further metrics?

More verbal details on exactly how all experiments were performed would be appreciated.

---

> ### Author Response · Authors · 2024-11-19
>
> >W1: Literature Overview: Why do you not cite works on DNNs and non-negative factor analysis in the interpretability framework? These are probably the closest works to your manuscript and constitute the core of your work. For example: https://proceedings.neurips.cc/paper_files/paper/2022/hash/e53280d73dd5389e820f4a6250365b0e-Abstract-Conference.html.
>
> Thank you very much for your constructive suggestions. After carefully reviewing the paper you provided, we found that L2I and BNDL indeed share some similarities.
> Specifically, both approaches leverage the idea of non-negative matrix factorization to interpret complex networks. However, due to their focus on different tasks, there are some differences in their specific implementation and areas of emphasis:
>
> (1) L2I primarily targets audio signal modeling, where non-negative matrix factorization needs to account for both time and frequency features. For instance, it integrates features from multiple intermediate layers of a deep network and performs upsampling and downsampling along both the time and frequency axes. In contrast, BNDL focuses on improving the network's uncertainty estimation capabilities by applying non-negative matrix factorization to the output features of the penultimate layer. These features are modeled as probability distributions, with specifically designed prior and posterior distributions to control the randomness of the variables.
>
> (2) L2I and BNDL both aim to increase the sparsity of variables in non-negative matrix factorization, but they differ in their learning algorithms. L2I enforces sparsity by adding an L1 norm term to the loss function, while BNDL introduces a Gamma prior distribution and uses KL divergence to ensure the sparsity of the features.
>
> Inspired by your suggestion, we believe it is important to include references to DNNs and non-negative matrix factorization within the interpretability framework in our paper.  **We have added some related references in line 68**, highlighted in blue.
>
> >W2: Theorem 1 is not a theorem (please check any statistical/ML literature, like AoS or NeurIPS for what is a theorem), neither is its "proof" is a proof, this is just a discussion. I would suggest you change the presentation.
>
> We appreciate your valuable feedback in helping us improve the clarity and appropriateness of our presentation. After thoroughly reviewing relevant materials and consulting Chapter 14, "Definitions, Theorems, and Proofs" from [1], we have updated Theorem 1 to Proposition and replaced "Proof" with "Analysis". Thank you again for pointing out this issue.
>
> [1] Houston K. Definitions, theorems and proofs. In: How to Think Like a Mathematician: A Companion to Undergraduate Mathematics. Cambridge University Press; 2009:99-102.
>
> >W3: The improvement of the performance does not seem to be significant at all, which is Ok, but the PAvPU might still seem questionable. How many receptions have been performed, what are your p-values? How, e.g., these numbers in Table 1 with +/- in front were calculated?
>
> Thank you for your insightful comments. We apologize for some missing experimental details in PAvPU, and we agree with your observation that BNDL does not lead to a substantial improvement in model performance. However, our primary goal is not to drastically improve performance, but to preserve the model's accuracy while enhancing its interpretability and uncertainty estimation capabilities.
> For the calculation of PAvPU, we sampled **20 times** to obtain 20 different logits, and then selected the top-2 classes based on their mean logits, performing a two-sample t-test to obtain the p-value. Following the approach used by CARD and MC Dropout, we set the p-value threshold to **0.05** to determine whether a sample is certain. Additionally, for each model, we performed 5 random runs, yielding five different PAvPU (0.05) and accuracy values. We then computed the mean and standard deviation of these values, as reported in Table 1.
>
> >W4: There are typos in the manuscript (which means it is human-written).
> We sincerely apologize for these errors. In the newly uploaded version, we have made the corrections. If you still notice any issues, please do not hesitate to let us know.

---

> > ### Author Response · Authors · 2024-11-19
> >
> > >Q1: The results on both uncertainty and disentanglement use one metric only each. Would it be possible include further metrics?
> >
> > Thank you for your suggestion. We have referred to previous literature on metrics for uncertainty estimation and added the Expected Calibration Error (ECE) [1], which evaluates the degree of miscalibration in a model. The ECE results for CIFAR-10 and CIFAR-100 are presented in the table below, where BNDL represents our proposed model, and the baseline for both datasets is ResNet18. Following the standard reporting convention for ECE [2], we have included the results of a single run for each experiment.
> >
> > |        ECE        | Cifar-10 | Cifar-100 |
> > |:-----------------:|:--------:|:---------:|
> > |   MC Dropout [3]  |   3.78   |   13.52   |
> > |       BM [2]      |   1.66   |    4.25   |
> > | Deep Ensemble [4] |   1.04   |    3.54   |
> > |        BNDL       | **0.40** |  **3.26** |
> >
> > It can be observed that BNDL achieves the best ECE results among these models, indicating that BNDL's predictive probabilities are well-aligned with its accuracy, thereby enhancing the calibration properties of neural networks.
> >
> > For the disentanglement measurement, after thoroughly reviewing the literature on disentanglement metrics, we found that most of these metrics, such as the Mutual Information Gap (MIG), are specifically designed for unsupervised disentangled representation learning tasks. These metrics often rely on disentanglement datasets (e.g., dSprites [5]), which provide labels for generative factors (e.g., shape, position, and size) but lack comprehensive labels required for classification tasks. Since BNDL operates under a supervised learning framework, it is currently challenging to directly apply such disentanglement metrics to evaluate our approach.
> >
> > We wonder if you might have any recommendations for disentanglement metrics that could be more suitable for evaluating BNDL. Your input would be highly appreciated, and thank you once again for your valuable feedback!
> >
> >
> > [1] Naeini, Mahdi Pakdaman, Gregory Cooper, and Milos Hauskrecht. "Obtaining well calibrated probabilities using bayesian binning." Proceedings of the AAAI conference on artificial intelligence. Vol. 29. No. 1. 2015.
> >
> > [2] Joo, Taejong, Uijung Chung, and Min-Gwan Seo. "Being bayesian about categorical probability." International conference on machine learning. PMLR, 2020.
> >
> > [3] Gal, Yarin, and Zoubin Ghahramani. "Dropout as a bayesian approximation: Representing model uncertainty in deep learning." international conference on machine learning. PMLR, 2016.
> >
> > [4] Lakshminarayanan, Balaji, Alexander Pritzel, and Charles Blundell. "Simple and scalable predictive uncertainty estimation using deep ensembles." Advances in neural information processing systems 30 (2017).
> >
> > [5] Matthey, Loic, et al. "dsprites: Disentanglement testing sprites dataset." May 2017.
> >
> > >Q2: More verbal details on exactly how all experiments were performed would be appreciated.
> >
> > Thank you for your suggestion. We have added additional experimental settings in the **appendix A.1 of the paper, with the changes highlighted in blue**. Also, we have uploaded the code in the supplementary materials. Please feel free to let us know if you have any further questions.

---

### Official Review · Reviewer_hX73 · 2024-11-03

**Soundness:** 2
**Presentation:** 2
**Contribution:** 2
**Rating:** 5
**Confidence:** 2

**Summary:**

The authors propose a Bayesian neural network where the final layer is modeled as a non-negative matrix factorization (NMF), i.e. $y \sim \Phi\Theta$, the motivation being that such a model would provide both predictive uncertainty estimates (because we learn a posterior distribution) and interpretability (because we learn a sparse factorization for $y$). Both $\Phi$ and $\Theta$ are modeled as Gamma distributions, and approximated variationally using the Weibull distribution. The authors evaluate their model (on accuracy, uncertainty, and sparsity) on CIFAR and ImageNet.

**Strengths:**

1. The methodology and description of the model and inference process is soundly written. Modeling as the factorization matrices as Gamma distributions should, in theory, encourage sparsity. The variational inference process that is described makes sense.

2. Section 4 is an interesting addition to the paper, which describes how the factorization matrices are (partially) identifiable under certain assumptions, and how the author's model satisfies such criteria.

3. I appreciate the sanity testing for uncertainty/accuracy correlation in Section 5.1.1.

**Weaknesses:**

My main criticisms would relate to the experimental results:

1. It is not clear to me why it is necessary to compare to non-Bayesian/point-estimate models, considering that the goal of the paper is to provide better uncertainty estimates. As such, I am not sure the ViT results are especially meaningful, i.e. it is unclear to me what I should be comparing ViT-BNDL to.

2. From Table 1, the PAvPU numbers for the ResNet model do not seem to be a huge improvement over the competing methods, especially the recent approaches (BM and CARD).

3. Is there a reason why sparsity values are not shown for the competing approaches in Section 5.1.2?

4. It is not clear to me that the interpretability evaluation metric in Section 5.2 is correct or useful. Specifically, why is unsupervised disentanglement important for ImageNet and CIFAR? Disentanglement does not imply interpretability, and disentangled features are not necessarily the correct ones to learn either (e.g. a spurious feature will be disentangled from a salient feature, but it doesn't mean we want to learn the former).

5. Relatedly, why do we not compare to competing approaches in Section 5.2? Would the authors be able to report the metric in Table 2 and the visualizations in Figure 4 for the models that BNDL was compared to earlier?

**Questions:**

The authors should respond to my questions in the Weaknesses section. I have no further questions, although I want to note that there are a fair number of typos in the paper that the authors should clean up.

E.g. I believe the first term inside the integral of Eq (4) should be $p(y|\Phi, \Theta)$?

---

> ### Author Response · Authors · 2024-11-19
>
> >W1: It is not clear to me why it is necessary to compare to non-Bayesian/point-estimate models, considering that the goal of the paper is to provide better uncertainty estimates. As such, I am not sure the ViT results are especially meaningful, i.e. it is unclear to me what I should be comparing ViT-BNDL to.
>
> Thank you for your insightful question, which gives us the opportunity to further clarify our motivation. We agree with your point that BNDL is a Bayesian model, and one of its objectives is to provide better uncertainty estimation for the model. However, after careful consideration, we still believe that comparing BNDL with non-Bayesian/point estimate models is necessary, and this is based on the following key reasons:
>
> (1) As a Bayesian decision layer module, we aim to demonstrate the **applicability of BNDL across various models**. To this end, we selected two representative models, ViT (a transformer-based model) and ResNet (a CNN-based model), as our baselines and incorporated the BNDL module into both.
>
> (2) We include both ResNet and ViT, two non-Bayesian models, in our comparison primarily to emphasize the following points: a) The incorporation of BNDL equips the baseline models with uncertainty estimation capabilities, without compromising accuracy. b) Models with the BNDL module exhibit better uncertainty estimation performance (as evidenced by PavPU) compared to other Bayesian uncertainty estimation methods.
>
> >W2: From Table 1, the PAvPU numbers for the ResNet model do not seem to be a huge improvement over the competing methods, especially the recent approaches (BM and CARD).
>
> Thank you for your comment. We agree with your point that the improvement in PAvPU on the ResNet model is not on a very large scale. However, we still believe that our enhancement in PAvPU is is meaningful rather than incremental. In addition to the numerical gains, where BNDL achieves consistent and stable improvements across three datasets, we would like to offer a deeper explanation by examining the underlying mechanism of PAvPU:
>
> Enhancing PAvPU is challenging, as it requires the model to have a robust rejection capability due to the inherent calculation mechanism of PAvPU. As shown in Table 1, for both BM and MC Dropout, the PAvPU on CIFAR-10 is even lower than the model's accuracy. Examining the PAvPU formula, $PAvPU = (n_{ac} + n_{iu}) / (n_{ac} + n_{au} + n_{iu} + n_{ic})$, we observe that accuracy can be similarly expressed as $(n_{ac} + n_{au}) / (n_{ac} + n_{au} + n_{iu} + n_{ic})$. For PAvPU to exceed accuracy, it requires $n_{iu}$ to be greater than $n_{au}$, meaning that the number of misclassified but uncertain samples must exceed the number of correctly classified but uncertain ones. This implies that the model must have an accuracy lower than 50\% on high-uncertainty samples, thus demonstrating a degree of rejection capability.
>
> Both BNDL and CARD consistently achieve PAvPU values higher than accuracy across all three datasets, with BNDL showing even greater improvement. For example, on ImageNet-1k, BNDL’s PAvPU improvement is seven times that of CARD, highlighting a stronger rejection capability. As illustrated in Figure 2, the testing accuracy on the two highest-uncertainty subsets of ImageNet-1k drops below 50\%, indicating that the model actively rejects uncertain samples during testing. This further underscores BNDL’s superior ability in uncertainty estimation.
>
> >W3: Is there a reason why sparsity values are not shown for the competing approaches in Section 5.1.2?
>
> Thank you for your question.  Following the comparison method used in [1],  we present the test accuracy of BNDL and Debuggable Networks [1] at the same levels of model sparsity. We believe this approach offers a more intuitive representation of how decreasing sparsity affects the model's test accuracy.  In figure 3 of section 5.1.2,  the orange line represents the method from [1], **with each point's horizontal coordinate indicating its respective sparsity value**. For ease of presentation, we use $1 - Sparsity$ ratio and apply a logarithmic scale to the horizontal axis in Figure 3. We have resubmitted our Supplementary Material and uploaded the data used to visualize Figure 3(b) from [1], including the Sparsity Values, in the file "ResNet-Sparse/vis/debuggable\_imagenet.txt".
>
> [1] Wong, Eric, Shibani Santurkar, and Aleksander Madry. "Leveraging sparse linear layers for debuggable deep networks." International Conference on Machine Learning. PMLR, 2021.

---

> ### Author Response · Authors · 2024-11-19
>
> >W4: It is not clear to me that the interpretability evaluation metric in Section 5.2 is correct or useful. Specifically, why is unsupervised disentanglement important for ImageNet and CIFAR?
> Disentanglement does not imply interpretability, and disentangled features are not necessarily the correct ones to learn either (e.g. a spurious feature will be disentangled from a salient feature, but it doesn't mean we want to learn the former).
>
> We sincerely thank you for dedicating your time and effort to reviewing our paper. We agree with your insightful observation that disentanglement does not inherently equate to interpretability.
> First, a primary challenge of visualization methods for interpretability lies in the multifaceted nature of neural activations: each neuron often responds to a wide variety of features. In this regard, disentangled representations aim to separate distinct, independent, and informative factors of variation within the data [1]. This separation can help address the multifaceted challenge and enhance the model’s interpretability. Specifically, as demonstrated in Figure 4, the top activated features of BNDL exhibit more consistent and semantically meaningful patterns, which are easier for humans to inspect and understand.
> Second, as you have pointed out, disentangled features may still include spurious features. While such features are independent of other dimensions, they might not be relevant to the task objective. We acknowledge this limitation and emphasize that correcting for spurious features is beyond the scope of this paper.
> Furthermore, we are actively exploring additional metrics to evaluate interpretability and would greatly appreciate your guidance or suggestions on selecting appropriate evaluation metrics.
>
> [1] Bengio et al. Representation learning: A review and new perspectives. IEEE Transactions on Pattern Analysis and Machine Intelligence, 35(8):1798–1828, 2013.
>
> >W5: Relatedly, why do we not compare to competing approaches in Section 5.2? Would the authors be able to report the metric in Table 2 and the visualizations in Figure 4 for the models that BNDL was compared to earlier?
>
> Thank you for your suggestion. Based on your feedback, we have added visualization results for BM [1], Debuggable Networks [2], and ResNet-50 [3] on ImageNet-1k, which can be found in **Figure 6 of the Appendix** in the revised submission. Please refer to this section for the updated results. Additionally, we have included a comparison of disentanglement results for BM [1], ResNet-50 [2], and BNDL on CIFAR-100, as presented in the table below (also updated in the Table 3 of the Appendix in the revised submission). Notably, BNDL consistently achieves the best disentanglement metrics, aligning with the conclusions drawn in Section 5.2 of the paper. This suggests that BNDL has successfully learned more identifiable features through the constraint of sparsity.
>
> | Cifar-100 |     SEPIN@1     |     SEPIN@10    |    SEPIN@100    |    SEPIN@1000   |    SEPIN@all    |
> |:---------:|:---------------:|:---------------:|:---------------:|:---------------:|:---------------:|
> |  ResNet18 |   3.17 ± 0.06   |   2.40 ± 0.03   |   1.79 ± 0.03   |   1.21 ± 0.01   |   1.21 ± 0.01   |
> |     BM    |   3.19 ± 0.05   |   2.53 ± 0.02   |   1.17 ± 0.02   |   1.28 ± 0.02   |   1.24 ± 0.01   |
> |    BNDL   | **3.91 ± 0.06** | **3.43 ± 0.03** | **2.77 ± 0.03** | **1.69 ± 0.03** | **1.69 ± 0.02** |
>
> [1] Joo, Taejong, Uijung Chung, and Min-Gwan Seo. "Being bayesian about categorical probability." International conference on machine learning. PMLR, 2020.
>
> [2] Wong, Eric, Shibani Santurkar, and Aleksander Madry. "Leveraging sparse linear layers for debuggable deep networks." International Conference on Machine Learning. PMLR, 2021.
>
> [3] He, Kaiming, et al. "Deep residual learning for image recognition." Proceedings of the IEEE conference on computer vision and pattern recognition. 2016.
>
> >Q1: typos in the paper
>
> Thank you for your thoughtful and detailed reviews. Regarding Equation 4, we respectfully maintain that $\theta$ and $\Phi$ should not appear on the left-hand side of the equation, as we marginalize over their probabilities. This approach aligns with conventions observed in [1], where Equation (1) directly expresses the probability as $p(x)$ rather than $p(x|z)$. Furthermore, we have addressed several issues in the manuscript, including typos, formula formatting, and overall clarity. We sincerely appreciate the time and effort you have dedicated to reviewing our paper.
>
> [1] Kingma, Diederik P. "Auto-encoding variational bayes." ICLR 2014.

---

> ### Comment · Reviewer_hX73 · 2024-12-03
> **Response to Authors**
>
> I thank the authors for their detailed response to my review. Overall I am satisfied with the responses/additional results to Points 2, 3, and 5, although I still think the core issues in Points 1 and 4 have not been addressed --- especially Point 4, which I think is the motivating issue of the paper. I think the authors have also misunderstood my point about Eq (4) in Q1. I think $\Phi$ and $\theta$ need to be conditioned on in the first term of the R.H.S., not that they need to appear on the L.H.S. Nevertheless, I thank the authors for taking the time to address each point with detail.

---

> ### Author Response · Authors · 2024-12-03
>
> Thank you for your response and for giving us the opportunity to clarify your concerns further.
>
> >Point 1
>
> Regarding Point 1, we would like to elaborate as follows:
> (1) Many prior works, such as [1], demonstrate the effectiveness of deep ensemble Bayesian models by comparing their accuracy (ACC) with point estimation models. This comparison highlights that Bayesian methods can not only achieve high accuracy but also provide more reliable uncertainty quantification. Similar comparisons have also been employed in [2] and [3].
> Furthermore, point estimation models can leverage softmax output probabilities to evaluate model confidence and assess calibration performance [4], achieving a certain degree of uncertainty quantification. Based on this reasoning, we believe that including the test accuracy of these point estimation models in the comparison is both fair and necessary.
> In addition to accuracy, we have included an uncertainty metric, Expected Calibration Error (ECE) [5], to provide a comprehensive evaluation. Below are the comparative results across different datasets:
>
> |        ECE        | Cifar-10 | Cifar-100 |
> |:-----------------:|:--------:|:---------:|
> |    ResNet18 [6]   |   3.82   |   13.48   |
> |   MC Dropout [2]  |   3.78   |   13.52   |
> |       BM [3]      |   1.66   |    4.25   |
> | Deep Ensemble [1] |   1.04   |    3.54   |
> |        BNDL       | **0.40** |  **3.26** |
>
> These results show that BNDL achieves the best ECE performance among these models, including the non-Bayesian ResNet18. This further demonstrates the advantage of Bayesian methods, as they introduce weight uncertainty to better address overfitting issues.
>
> (2) The purpose of our work is to achieve both uncertainty estimation and enhanced interpretability while maintaining or even improving model accuracy. To illustrate that BNDL does not compromise the performance of non-Bayesian models in terms of test accuracy, we included comparisons with baseline non-Bayesian models. For example, we presented ViT-BNDL results to demonstrate that it neither degrades nor slightly improves the performance of ViT-Base. This approach aligns with the methodologies adopted in [1-3].
>
> > Point 4
>
> Regarding Point 4, while we agree with your perspective that disentanglement does not inherently equate to interpretability, we respectfully assert that disentanglement enhances model interpretability by reducing variable ambiguity and redundancy. This connection has been substantiated through empirical validation in numerous studies [7, 8, 16, 17, 18]. For instance, [7] emphasizes that disentangled representations isolate distinct factors of variation in data, yielding representations that are not only more interpretable but also predictive. Similarly, [8] highlights the ability of disentangled representations to enhance interpretability by effectively separating independent factors of variation, leading to semantically meaningful and clearer insights.
>
> In light of this relationship, many studies employ disentanglement to demonstrate the interpretability of models [9–12]. For example, [9] demonstrates a positive correlation between disentanglement and semantic interpretability, using disentanglement metrics for quantitative evaluation. Furthermore, [10] showcases the use of disentanglement results on the Sprites dataset to illustrate interpretability. Related methodologies are also found in [11] and [12]. Specifically, [11] underscores how sparse disentanglement reduces model complexity, making model behavior easier to understand, while [12] demonstrates that disentanglement induced by non-negativity aids in generating interpretable representations, quantified using the SEPIN metric.
>
> Notably, many of these works perform disentanglement experiments in unsupervised settings, as seen in [8, 12, 16]. Moreover, studies such as [13–15] have conducted experiments within the evaluation framework of unsupervised disentangled representation learning. This approach is particularly valuable as unsupervised disentanglement enables the interpretation of latent causal factors within image data without explicit labels, leveraging disentangled features to improve the model's performance on downstream tasks.
>
> We hope these further clarifications adequately address your concerns regarding Point 4.
>
> >About Equation 4
>
> Regarding Equation 4, we thank your further clarification and we apologize for misunderstanding your point. If there is an opportunity to revise the PDF, we will update Equation 4 accordingly.

---

> ### Author Response · Authors · 2024-12-03
>
> ### References
>
> [1] Lakshminarayanan, B., Pritzel, A., \& Blundell, C. (2017). Simple and Scalable Predictive Uncertainty Estimation using Deep Ensembles. In: Neural Information Processing Systems (NeurIPS).
>
> [2] Gal, Yarin, and Zoubin Ghahramani. "Dropout as a bayesian approximation: Representing model uncertainty in deep learning." international conference on machine learning. PMLR, 2016.
>
> [3] Joo, Taejong, Uijung Chung, and Min-Gwan Seo. "Being bayesian about categorical probability." International conference on machine learning. PMLR, 2020.
>
> [4] Guo, C., Pleiss, G., Sun, Y., & Weinberger, K. Q. (2017). On Calibration of Modern Neural Networks. In: Proceedings of the 34th International Conference on Machine Learning (ICML).
>
> [5] Naeini, Mahdi Pakdaman, Gregory Cooper, and Milos Hauskrecht. "Obtaining well calibrated probabilities using bayesian binning." Proceedings of the AAAI conference on artificial intelligence. Vol. 29. No. 1. 2015.
>
> [6] He, K., et al. (2016). Deep residual learning for image recognition. CVPR.
>
> [7] Bengio, Yoshua, Aaron Courville, and Pascal Vincent. "Representation learning: A review and new perspectives." IEEE transactions on pattern analysis and machine intelligence 35.8 (2013): 1798-1828.
>
> [8] Higgins, Irina, et al. "beta-vae: Learning basic visual concepts with a constrained variational framework." ICLR (Poster) 3 (2017).
>
> [9] Eastwood, Cian, and Christopher KI Williams. "A framework for the quantitative evaluation of disentangled representations." 6th International Conference on Learning Representations. 2018.
>
> [10] Hu, Qiyang, et al. "Disentangling factors of variation by mixing them." Proceedings of the IEEE Conference on Computer Vision and Pattern Recognition. 2018.
>
> [11] Lachapelle, Sébastien, et al. "Synergies between disentanglement and sparsity: Generalization and identifiability in multi-task learning." International Conference on Machine Learning. PMLR, 2023.
>
> [12] Wang, Yifei, et al. "Non-negative Contrastive Learning." The Twelfth International Conference on Learning Representations.
>
> [13] Ruiz, Adrià, et al. "Learning Disentangled Representations with Reference-Based Variational Autoencoders." ICLR workshop on Learning from Limited Labeled Data. 2019.
>
> [14] Hsu, Wei-Ning, Yu Zhang, and James Glass. "Unsupervised learning of disentangled and interpretable representations from sequential data." Advances in neural information processing systems 30 (2017).
>
> [15] Chen, Xi, et al. "Infogan: Interpretable representation learning by information maximizing generative adversarial nets." Advances in neural information processing systems 29 (2016).
>
> [16] Kim, Hyunjik, and Andriy Mnih. "Disentangling by factorising." International conference on machine learning. PMLR, 2018.
>
> [17] Ridgeway, Karl, and Michael C. Mozer. "Learning deep disentangled embeddings with the f-statistic loss." Advances in neural information processing systems 31 (2018).
>
> [18] Esmaeili, Babak, et al. "Structured disentangled representations." The 22nd International Conference on Artificial Intelligence and Statistics. PMLR, 2019.

---

### Official Review · Reviewer_4YUN · 2024-11-04

**Soundness:** 3
**Presentation:** 3
**Contribution:** 3
**Rating:** 6
**Confidence:** 3

**Summary:**

The authors focus on improving the interpretability of deterministic neural net, by inserting a probabilistic layer that provides a non-negative factorization for an interpretable classification.
They prove (partial) identifiability and evaluate the method on several image classification benchmarks.

**Strengths:**

- The method is generic enough to be added to an arbitrary deep architecture
- It performs well under varying experimental settings and architectures and can keep/improve upon its non-interpretable counterpart while greatly improving interpretability


One caveat that should be noted is that I am not too familiar with the current state of the art in interpretability research. As presented, the results look significant, but they might not be.

**Weaknesses:**

- The experiments are limited to a small set of interpretability baselines.
- Sec 3.1 "We first adopt a Bayesian perspective to re-examine the DNNs". This framing is rather crude. The vague fact that one could interpret the input to a softmax as a delta distribution over a latent variable alone is not enough to call something Bayesian. A Bayesian approach requires a well-specified prior combined with a posterior inference. Simply making a model (indirectly) probabilistic is not Bayesian.
- l218 "it lacks reparameterization and cannot be optimized"
You can always rely on what is often known as the REINFORCE approach, i.e., $\nabla_a E_{x\sim q_a(x)}[f(x)] = E_{x \sim q_a(x)}[f(x)\nabla_a \log q_a(x)]$. However, you usually don't want to do this as it will punish your gradients with a huge variance, which is why your proposal is much more stable and sane. But as long as you have a density you could in theory do it.
- (12) the left hand side should be p(Y|X), as you marginalize on the rhs over $\theta$ and $\Phi$
- Given the Bayesian framing of the paper a short discussion or mention of what is known as last-layer BNNs is missing. These combine a deterministic network trunk with a Bayesian inference over the last layer of a neural net. See, e.g., the references in Harrison et al. (2024). (This is not necessarily the best reference for this research direction that has been growing recently, but one whose references can serve as guidance for a more generic reference.)
- In a similar direction goes the field of evidential deep learning also known as prior networks, where a prior to the last layer is inserted in a different way. See, e.g., Sensoy et al. (2018) for classification or Amini et al. (2020) and Malinin et al. (2020) for regression. Both research directions, i.e., EDL and LL-BNNs have a different aim than the authors' proposal but rely on similar mechanics.
- Regarding overconfidence in l167, I would have expected a reference to the first main study in that direction by Guo et al. (2017). (At least to my knowledge.)
- In Thm 1 $e_{(k)}$ is not introduced

_____
_Amini et al., Deep Evidential Regression (2020)_
_Guo et al., On Calibration of Modern Neural Networks (2017)_
_Harrison et al., Variational Bayesian Last Layers (2024)_
_Malinin et al., Regression prior networks (2020)_
_Sensoy et al., Evidential Deep Learning to Quantify Classification Uncertainty (2018)_





### Typos
The paper contains a lot of typos and missing articles, which should be fixed in a thorough proofreading round. A subset of these are
- l81 Furthermore, we provide
- l105,l107 (and maybe others) the citation style is broken use \citep and \citet correctly, please follow the style guide
- most equations, e.g., (1), (2) lack proper punctuation
- l206 $\mathbb{R}_+$
- l241 "where $h_j$ is an extracted feature
- l272 of the log-likelihood
- l345 are described in
- l440 misclassification, e.g., in the
- A lot of references are broken, e.g., Dosovitskiy et al. (2020) is a published paper, so is Kingma & Ba's Adam etc.

**Questions:**

- Q1: The authors provide a theoretical complexity analysis. What is the practical runtime increase compared to a simple fully-connected layer?
- Q2: Can the authors provide greater detail on their relatio nto Wang et al. (2024)? There is a strong relation in the method (NMF) and aim to improve interpretability, yet within this paper it is only ever mentioned in passing without being fully introduced nor compared against. The same holds, e.g., for Duan et al. (2024).

---

> ### Author Response · Authors · 2024-11-19
>
> We sincerely thank you for your valuable suggestions on our paper. Your feedback has been instrumental, not only in helping us correct certain imprecise expressions but also in offering constructive and detailed recommendations for enhancing the paper’s overall framework. Based on your insights, we have revised the resubmitted version, focusing on the following key improvements:
>
> (1) We have refined imprecise expressions from the original manuscript, including referring to the reformulated form of classical DNNs as a latent variable model, rephrasing the Weibull posterior estimation, adding a citation to Guo et al. (line 174) in the discussion on overconfidence, and correcting several typographical errors.
>
> (2) Following your suggestion, we have added a discussion on BLL and EDL in **Sec. 2.1 in our manuscript**. Additionally, we have expanded the interpretability experiments by including comparative baselines, as detailed in the newly uploaded version, specifically in **Figure 6 and Table 3 in the Appendix A.3**. These updates have been highlighted in blue. We kindly invite you to refer to this section for further details.
>
> For other areas needing additional clarification, we outline them as follows:
>
> >W1: The experiments are limited to a small set of interpretability baselines.
>
> Thank you for your valuable suggestions. In response, we have incorporated comparative visualization results for BM [1] and Debuggable Networks [2] within our interpretability experiments. **Please see Figure 6 in the appendix** of the updated version for further details. Additionally, we have included a comparison of disentanglement metrics between BM and BNDL on CIFAR-100, which can be found in **Table 3 of the appendix**. If you have any additional feedback or suggestions, please don’t hesitate to share.
>
> [1] Joo, Taejong, Uijung Chung, and Min-Gwan Seo. "Being bayesian about categorical probability." International conference on machine learning. PMLR, 2020.
>
> [2] Wong, Eric, Shibani Santurkar, and Aleksander Madry. "Leveraging sparse linear layers for debuggable deep networks." International Conference on Machine Learning. PMLR, 2021.
>
> >W2 and W3:  Sec 3.1 "We first adopt a Bayesian perspective to re-examine the DNNs";  l218 "it lacks reparameterization and cannot be optimized"
>
> We apologize for the imprecise expression and sincerely thank you for your corrections, which are critical to improving the quality of our paper. First, we agree with you that interpreting the input to a softmax as a delta distribution over a latent variable is not sufficient to classify the model as a Bayesian model. In line with the expressions in [1, 2], we have updated "Bayesian perspective" to "latent variable model". Secondly, We agree that the reinforcement approach can indeed be used to optimize non-differentiable objective functions [3], and it indeed frequently suffers from high variance and convergence difficulties.
> In response to your suggestions and to improve paper's quality, **we have revised the phrasing of the Weibull posterior estimation in Section 3.3**, with the changes highlighted in blue.
>
> [1] Dhuliawala, Shehzaad et al. “Variational Classification.” Trans. Mach. Learn. Res. 2024 (2023): n. pag.
>
> [2] Joo, Taejong et al. “Being Bayesian about Categorical Probability.” International Conference on Machine Learning (2020).
>
> [3] Williams, R. J. (1992). "Simple statistical gradient-following algorithms for connectionist reinforcement learning." Machine Learning, 8(3-4), 229-256.
>
> >W5 and W6: Given the Bayesian framing of the paper a short discussion or mention of what is known as last-layer BNNs is missing. These combine a deterministic network trunk with a Bayesian inference over the last layer of a neural net. In a similar direction goes the field of evidential deep learning also known as prior networks, where a prior to the last layer is inserted in a different way. Both research directions, i.e., EDL and LL-BNNs have a different aim than the authors' proposal but rely on similar mechanics.
>
> Thank you very much for your valuable suggestions. After carefully reading the articles you provided, we found them really valuable for completing the structure of our paper. Consequently, we have incorporated a discussion on EDL and LL-BNNs into the latest version of our manuscript, **highlighted in blue in the Secion 2.1**. Please refer to this section for more details.
>
> >W4, W7 and W8: (12) the left hand side should be p(Y|X) ; Regarding overconfidence in l167, I would have expected a reference to the first main study in that direction by Guo et al. (2017). (At least to my knowledge.);In Thm 1 $e_{(k)}$ is not introduced.
>
> Thank you for your detailed feedback. We have made revisions to Equation 12 and added a citation to Guo et al. (2017) in line 167. In Section 4, $e_{(k)}^T$ represents the $k$-th standard row vector in vector space, and we have added the corresponding revision in Sec.4, highlighted in blue.

---

> > ### Comment · Reviewer_4YUN · 2024-11-19
> >
> > - Regarding the new BLL discussion.
> >
> > The current discussion is insufficient. My aim in providing the Harrison et al. reference was not to get you to spam random Harrison references. The two 2020 papers are, as one could guess from the title, meta-learning papers rather than BLL papers. Wilson et al. (2016) is a kernel learning paper, and thus also not the same. Please provide a proper overview of the related work. Neither is Joo et al. (2020) in the strict sense a BLL paper contrary to your claim in the appendix.
> >
> > - Regarding the new EDL discussion.
> >
> > What do you mean with _"such as modeling ”evidence” vectors with task-specific priors (Sensoy et al., 2018; Malinin et al., 2020), or directlylearning task-specific parameter distributions (Amini et al., 2020)"_? What is a task-specific prior vs a task-specific parameter distribution?
> >
> >
> > - **Additionally, we have expanded the interpretability experiments by including comparative baselines, as detailed in the newly uploaded version, specifically in Figure 6 and Table 3 in the Appendix A.3**
> >
> > If I see it correctly these are just extensions of the current baselines. There are no new methods you compared against, are there?
> >
> >
> > - ** In response to your suggestions and to improve paper's quality, we have revised the phrasing of the Weibull posterior estimation in Section 3.3, with the changes highlighted in blue.**
> >
> > Very minor comment here. It is not difficult to reparameterize it, it is simply difficult to learn anything useful due to the noisy signal.

---

> ### Author Response · Authors · 2024-11-19
>
> >Q1: The authors provide a theoretical complexity analysis. What is the practical runtime increase compared to a simple fully-connected layer?
>
> Thank you for your question. To further validate our theoretical complexity analysis, we provide a comparison of the runtime between BNDL and a single fully connected (FC) layer on CIFAR-100, where the experimental setup remains consistent with the configuration used in Table 1 for CIFAR-100. In the table, we present the training and testing runtimes of the single-layer decision layer, as well as the training and testing times for the entire network (i.e., whole ResNet18).
>
> | **Cifar-100** | single layer batch training time (s) | single layer batch testing time (s) | Network epoch training time (s) | Network epoch testing time (s) |
> |:-------------:|:------------------------------------:|:-----------------------------------:|:-------------------------------:|:------------------------------:|
> |      BNDL     |                $9.16 \times 10^{-4}$               |               $1.43 \times 10^{-4}$               |                38               |                2               |
> |    Fc-layer   |                $1.06 \times 10^{-4}$               |               $9.02 \times 10^{-4}$               |                37               |                2               |
>
> From the table, we can draw the following two observations:
>
> (1) We acknowledge that the runtime of BNDL is indeed higher than that of a standard FC layer.
> During training, the reparameterization of both $\theta$ and $\Phi$ introduces unavoidable computational overhead. However, compared to the overall training time of the model, the additional computational time introduced by BNDL is negligible. As shown in the table above, it can be seen that the training time for the model using BNDL is only 1 second longer than that of the standard FC layer, representing an increase of approximately 2.7%;
>
>
> (2) During testing, since only the mean of the learned distribution is utilized, the runtime is significantly reduced. The testing time is only $5.28 \times 10^{-5}$ seconds longer than that of the simple FC layer, and the testing times for both models are comparable.
> Overall, the experiments result can further supports the conclusion derived from our theoretical analysis: **the added time complexity of the proposed BNDL is almost negligible compared to the overall time complexity of ResNet.**

---

> > ### Author Response · Authors · 2024-11-19
> >
> > >Q2: Can the authors provide greater detail on their relation to Wang et al. (2024)? There is a strong relation in the method (NMF) and aim to improve interpretability, yet within this paper it is only ever mentioned in passing without being fully introduced nor compared against. The same holds, e.g., for Duan et al. (2024).
> >
> > Thank you for your question. We apologize for not providing a detailed explanation of the relationship between NMF and interpretability.
> > Specifically, the non-negativity constraint in NMF ensures that the decomposition of the data matrix can be interpreted as an "additive combination" rather than a complex mathematical operation involving cancellations between positive and negative terms. This characteristic aligns more closely with how humans naturally understand patterns in real-world data.
> > Therefore, both BNDL and [1] leverage the principles of NMF by modeling the final layer of the network as an NMF problem, employing non-negativity constraints to enhance interpretability.
> > However, there remain key differences between BNDL and [1], which include the following:
> >
> > (1) NCL remains a point-estimation model that enforces non-negativity constraints by applying a ReLU activation function to the features. In contrast, BNDL probabilistically models the features as non-negative distributions, which enables uncertainty estimation—something that NCL does not accommodate.
> >
> > (2) Although NMF naturally provides some degree of disentanglement, it is often insufficient for handling complex data [2], necessitating additional constraints to improve this effect. Unlike NCL, which does not introduce such extra constraints, BNDL applies a Gamma prior to the factors in matrix decomposition, further enhancing the sparsity and non-linearity of these factors. This additional constraint strengthens both the disentanglement and interpretability of the model.
> >
> > Regarding [3], it also recognizes the additive property introduced by non-negativity, which aids in the disentanglement of the network. It leverages the non-negativity and sparsity of the Gamma distribution to design a Variational Autoencoder (VAE) generative model, achieving better disentanglement performance compared to Gaussian-VAE. In general, while BNDL shares some similar tools with [1] and [3], there are notable differences in their network designs and architectures, as well as the distinct objectives they serve in different tasks.
> >
> > [1] Wang, Yifei, et al. "Non-negative Contrastive Learning." The Twelfth International Conference on Learning Representations.
> >
> > [2] Hoyer, Patrik O. "Non-negative matrix factorization with sparseness constraints." Journal of machine learning research 5.9 (2004).
> >
> > [3] Duan, Zhibin, et al. "A Non-negative VAE: the Generalized Gamma Belief Network." arXiv preprint arXiv:2408.03388 (2024).

---

> > > ### Comment · Reviewer_4YUN · 2024-11-19
> > >
> > > Is this discussion on their relationship also included in the revised version of the paper? I couldn't find it.
> > >
> > > Aside from that, Wang et al. seem to be sufficiently flexible to provide some disentanglement results and claims identifiability to a certain extend. Note that you provided a reference to a linear NMF paper.
> > > How would a deterministic NMF paper compare to your probabilistic one for the non-uncertainty-based experiments? (Accuracy, disentanglement, sparsity,...)

---

> > ### Comment · Reviewer_4YUN · 2024-11-19
> >
> > Thanks for the new runtime events. They look nice and give further strength to your claim of BNDL being a practicable method.

---

> ### Author Response · Authors · 2024-11-21
>
> > Regarding the new BLL discussion.
>
> We sincerely apologize for not fully understanding your suggestions earlier. We realize that our discussion should focus more on literature closely aligned with the core principles of Bayesian Last Layer (BLL), rather than merely referencing papers employing similar techniques. To address this, we have re-examined the literature and identified some following representative works, for instance:
>     [1] applies Bayesian linear regression to the outputs of the final hidden layer to effectively mitigate overfitting.
>     [2] introduces prior information into the objective function to enhance feature smoothness and diversity, thereby improving predictive uncertainty.
>     [3] employs Laplace Approximations (LA) on the linear layer of the last layer in deep networks, approximating posterior distributions to achieve cost-effective yet reliable probabilistic predictions.
> Based on the newly reviewed literature [1-5], we plan to revise the discussion in Section 2.1 of our manuscript and have updated our submission accordingly, highlighted in magenta. We look forward to receiving your further guidance. Additionally, we agree with your point that, strictly speaking, Joo et al. (2020) is not a typical BLL study, as it does not make changes to the last linear layer. We have addressed this by revising the imprecise statements in the appendix.
>
>
> [1] Ober, Sebastian W., and Carl E. Rasmussen. "Benchmarking the Neural Linear Model for Regression." Second Symposium on Advances in Approximate Bayesian Inference.
>
> [2] Watson, Joe, et al. "Latent derivative Bayesian last layer networks." International Conference on Artificial Intelligence and Statistics. PMLR, 2021.
>
> [3] Daxberger, Erik, et al. "Laplace redux-effortless bayesian deep learning." Advances in Neural Information Processing Systems 34 (2021): 20089-20103.
>
> [4] Harrison, James, John Willes, and Jasper Snoek. "Variational Bayesian Last Layers." The Twelfth International Conference on Learning Representations.
>
> [5] Weber, Noah, et al. "Optimizing over a bayesian last layer." NeurIPS workshop on Bayesian Deep Learning. 2018.
>
>
> >Regarding the new EDL discussion.
>
> We are sorry about the confusion we caused. In the first sentence, the term "task-specific prior" was intended to distinguish between the works by Sensoy et al. (2018) and Malinin et al. (2020), which apply to classification and regression tasks, respectively, where the former placing a Dirichlet distribution on class probabilities and the latter introducing a Normal-Wishart distribution for regression.
> In the second sentence, the phrase "task-specific parameter" was meant to emphasize Amini et al. (2020), which directly infers the hyperparameters of evidential priors, unlike Sensoy et al. (2018), which relies on regularizing divergence to a fixed, predefined prior.
> We sincerely apologize for the lack of clarity in our previous wording again. We have now revised the discussion on EDL in the main text and welcome your valuable feedback to further refine our work.
>
> >There are no new methods you compared against, are there?
>
> Allow us to confirm your suggestion: do you mean that we should select some new explainability methods to conduct the experiments in Table 1, or that we should explore additional post-hoc explanation methods, beyond LIME, to visualize BNDL? If it is the latter, we have conducted a new set of experiments, in which we added GradCAM [1] for explaining our model. GradCAM generates heatmaps by leveraging the relationship between feature maps at specific layers of the neural network and the final prediction of the classification task. You can refer to Figure 7 in the Appendix of the newly uploaded version for detailed results. If we have misunderstood, please let us know, and we would be happy to discuss further.
>
> [1] Selvaraju, Ramprasaath R., et al. "Grad-cam: Visual explanations from deep networks via gradient-based localization." Proceedings of the IEEE international conference on computer vision. 2017.
>
> >it is simply difficult to learn anything useful due to the noisy signal.
>
> Thank you for your valuable feedback, we have revised this phrasing in line 237,

---

> ### Author Response · Authors · 2024-11-21
>
> >Aside from that, Wang et al. seem to be sufficiently flexible to provide some disentanglement results and claims identifiability to a certain extend. Note that you provided a reference to a linear NMF paper. How would a deterministic NMF paper compare to your probabilistic one for the non-uncertainty-based experiments? (Accuracy, disentanglement, sparsity,...)
>
> We agree with your perspective that their framework indeed provides a flexible approach to unsupervised disentangled representation learning.
> On the other hand, following your suggestion, we incorporated NMF into a supervised learning framework and conducted experiments on CIFAR-100. Consistent with the experimental settings in the paper, ResNet-NMF adopts the ResNet18 baseline model. The experiments were performed on an NVIDIA 3090 GPU. We ran five experiments with different random seeds and computed the mean and variance of the results. The experimental outcomes are presented in the table below.
>
> |   Cifar-100   |     Accuracy (%)     |   1 - Sparsity  |
> |:-------------:|:----------------:|:---------------:|
> |    ResNet18   |   74.62 ± 0.23   |   0.97 ± 0.02   |
> |  ResNet18-NMF |   76.73 ± 0.16   |   0.23 ± 0.01   |
> | ResNet18-BNDL | **79.82 ± 0.13** | **0.12 ± 0.01** |
>
>
> |   Cifar-100   |     SEPIN@1     |     SEPIN@10    |    SEPIN@100    |    SEPIN@1000   |    SEPIN@all    |
> |:-------------:|:---------------:|:---------------:|:---------------:|:---------------:|:---------------:|
> |    ResNet18   |   3.17 ± 0.06   |   2.40 ± 0.03   |   1.79 ± 0.03   |   1.21 ± 0.01   |   1.21 ± 0.01   |
> |  ResNet18-NMF |   3.21 ± 0.03   |   2.95 ± 0.02   |   1.87 ± 0.03   |   1.33 ± 0.02   |   1.23 ± 0.03   |
> | ResNet18-BNDL | **3.91 ± 0.06** | **3.43 ± 0.03** | **2.77 ± 0.03** | **1.69 ± 0.03** | **1.69 ± 0.02** |
>
> From the experiments on accuracy and disentanglement, the following observations can be made:
> (1) Incorporating NMF into the network enhances its performance. By introducing non-negativity constraints on both weights and features, ResNet-NMF achieves more sparse weights and improved disentanglement performance.
>
> (2) BNDL consistently achieves the best results on the CIFAR-100 dataset. This can be attributed to its probabilistic modeling and the additional sparsity and nonlinear constraints introduced by the Gamma prior, leading to greater weight sparsity and further improved disentanglement.
>
> Due to the page limit, We now include these experimental results and discussions on related work in the revised version's Appendix A.3, highlighted in magenta, and point it out in the main text (l284).

---

### Official Review · Reviewer_Wv5o · 2024-11-04

**Soundness:** 4
**Presentation:** 2
**Contribution:** 3
**Rating:** 8
**Confidence:** 3

**Summary:**

The manuscript introduces a Bayesian Nonnegative Decision Layer (BNDL) for deep neural network classifiers, with the intent of reformulating them as a factor analysis. This is shown to enhance the interpretability and uncertainty-estimation capabilities of the networks, at least on the examined datasets.

**Strengths:**

The idea is rigorous and the implementation introduces a minimal overhead over an existing network.

**Weaknesses:**

The presentation should be improved, see questions

**Questions:**

- How do we know the distinction between epistemic and aleatoric uncertainty? The epistemic uncertainty should go to zero for large data limit. Is this the case in the present modeling?
- In Sec. "uncertainty evaluation metric", how are the various n_ac,  n_au, n_ic, n_iu actually computed?
- Eq. 9: what is the difference between f_NNand f_lambda?
- in section 3.2, the switch from Gaussian to Gamma distribution is unclear. Is the Gaussian distribution used in this work? It seems not but the sentence "Both θ and Φ are sampled from a Gaussian distribution" points otherwise.
- in sparsity measurement a threshold of 10^-5 is defined for the weights. Shouldn't this come from the analysis of the distribution of weights, rather than just providing a number?
-in table 1, it seems the application of BNDL to ResNet introduces a significantly better improvement than when applied to ViT. Can the reason of this be understood?

Typos:
- row 178, "uncertainty-refers" should not have a hyphen
- 282 Killback–Leibler
- 345 "descirbed"
- 355 "we uses"
- 371 "Perforamce"

---

> ### Author Response · Authors · 2024-11-19
>
> We sincerely appreciate the time and effort you invested in reviewing our paper, as well as your valuable feedback.
> In response to your suggestions, we have first corrected several typo errors in the manuscript.
> Below, we address each of your comments individually.
>
> > Q1: How do we know the distinction between epistemic and aleatoric uncertainty? The epistemic uncertainty should go to zero for large data limits. Is this the case in the present modeling?
>
> Thank you for your insightful question, which motivates us to think further about the proposed model.
> Specifically, aleatoric uncertainty refers to the inherent uncertainty in the data itself, such as measurement noise or the randomness of the data, which is independent of the model and typically cannot be eliminated by increasing the amount of data. Epistemic uncertainty arises from a lack of knowledge about the true mapping from inputs to outputs and can usually be reduced by increasing the training data. In BNDL, since we use a Weibull distribution to approximate the posterior distribution, epistemic uncertainty is reflected in the shape and scale parameters of the Weibull distribution.
> As the training data increases, the shape and scale parameters govern the probability density function of the Weibull posterior distribution, leading it to become more concentrated and exhibit reduced variance.
>
>
> #### For Q2-Q4, we apologize for the lack of clarity on some points in the paper. Next, let us clarify your concerns.
>
> > Q2: In Sec. "uncertainty evaluation metric", how are the various n_ac, n_au, n_ic, n_iu actually computed?
>
>  First, we obtain the accurate and inaccurate classified samples based on the model's results on the dataset. The next key step is to distinguish between certain and uncertain samples. Intuitively, we set the number of samples as $M$, perform $M$ inference runs and obtain $M$ sets of logits from the model, denoted as $u$, where $u \in \mathbb{R}^{N \times C \times M}$. Here, $N$ is the number of samples, and $C$ is the number of classes.
> We then take the average of these $M$ logits and select the top-2 class indices to retain. For the top-2 logits obtained for each class, we perform a two-sample t-test to calculate the p-value.
> We use this resulting p-value to assess the uncertainty of each sample, following the approach in prior work [1]. Specifically, samples with a p-value greater than 0.05 are considered certain, while those with a p-value less than 0.05 are regarded as uncertain. This method has been adopted in multiple previous studies [2, 3] and has proven effective in measuring uncertainty.
>
> [1] Fan, Xinjie, et al. "Contextual dropout: An efficient sample-dependent dropout module." arXiv preprint arXiv:2103.04181 (2021).
>
> [2] Mukhoti, Jishnu, and Yarin Gal. "Evaluating bayesian deep learning methods for semantic segmentation." arXiv preprint arXiv:1811.12709 (2018).
>
> [3] Mukhoti, Jishnu, et al. "Deep deterministic uncertainty: A new simple baseline." Proceedings of the IEEE/CVF Conference on Computer Vision and Pattern Recognition. 2023.
>
> >Q3: Eq. 9: what is the difference between f_NNand f_lambda?
>
> In Eq.9, $f_{NN}$ and $f_{\lambda}$ represent two distinct networks. Specifically, $f_{NN}$ acts as a feature extractor, encompassing all layers from the input layer to the penultimate layer, tasked with extracting increasingly abstract features from the input data. Meanwhile, $f_{\lambda}$ is a neural network used to obtain the scale parameter $\lambda$ for the Weibull distribution, alongside a corresponding neural network $f_{k}$ for obtaining the shape parameter $k$. In response to your question, **we have made modifications in Sec 3.3 of the paper, with changes highlighted in blue.**
>
> >Q4: in section 3.2, the switch from Gaussian to Gamma distribution is unclear. Is the Gaussian distribution used in this work? It seems not but the sentence "Both θ and Φ are sampled from a Gaussian distribution" points otherwise.
>
> We apologize again for any confusion caused. In Eq. 4, we reformulate the classical DNN as a generative model.  As noted in line 198 (under Eq. 4), a common approach to improving a DNN into a Bayesian model is to model its parameters using a Gaussian distribution.  However, Gaussian distributions lack non-negativity and sparsity properties, which hinder the network's ability to learn sparse, disentangled features.  To address this limitation, BNDL models $\theta$ and $\Phi$ as Gamma distributions, leveraging their non-negative and sparse characteristics, as shown in Eq. 5.  **In response to your suggestion, we have made revisions at line 197, highlighted in blue.**

---

> > ### Author Response · Authors · 2024-11-19
> >
> > >Q5: in sparsity measurement a threshold of 10^-5 is defined for the weights. Shouldn't this come from the analysis of the distribution of weights, rather than just providing a number? -in table 1, it seems the application of BNDL to ResNet introduces a significantly better improvement than when applied to ViT. Can the reason of this be understood?
> >
> > Thank you for your insightful questions again.
> > For the first question, in order to ensure a fair comparison in the experiment of Figure 3, the sparsity measurement in BNDL is consistent with the definition of sparsity in [1], where a threshold of $1 \times 10^{-5}$ is used. We acknowledge that adjusting the threshold dynamically based on the distribution of weights could be a more reasonable approach for measuring sparsity. Following your suggestion, we have analyzed the weight distribution on the ImageNet-1k dataset, using weights from ResNet-BNDL in Table 1, and obtained the following table.
> >
> >
> > |   weight  |  w >= 0 | w >= 1e-5 | w >= 1e-4 | w >= 1e-3 | w >= 1e-2 | w >= 1e-1 | w >= 0.2 | w >= 0.3 |
> > |:---------:|:-------:|:---------:|:---------:|:---------:|:---------:|:---------:|:--------:|:--------:|
> > |  BNDL(\%)  | 100.000 |   4.010   |   4.006   |   3.987   |   3.867   |   2.007   |   0.712  |   0.227  |
> > | ResNet(\%) | 100.000 |   99.971  |   99.695  |   96.907  |   69.456  |   1.800   |   0.162  |   0.015  |
> >
> > It can be observed that weights exceeding the threshold of $1 \times 10^{-5}$ account for approximately 4\% of the total, indicating that this threshold effectively captures the significant portion of non-zero weights in the network.This observation aligns with the sparsity induced by the Gamma prior distribution introduced in BNDL, where the probability density function of Gamma(1,1) exhibits a high density near zero. Additionally, BNDL exhibits a higher proportion of larger parameter values (e.g., those greater than 0.3) compared to the base model. This indicates that the model relies more heavily on certain critical parameters to accomplish the task, further supporting and validating our motivation.
> >
> > Regarding the second question, as shown in Table 1, incorporating BNDL consistently enhances the performance of the baseline models, with particularly notable improvements in the ResNet-based framework.  This difference likely stems from variations in training methodologies and model characteristics. Specifically, the ViT/ViT-BNDL models leverage pre-trained weights that have undergone extensive training on the ImageNet-21k dataset. This pre-training stage significantly mitigates epistemic uncertainty in the ViT-Base model due to the dataset's broad coverage.
> > In contrast, the ResNet model used in our experiments has not been pre-trained on large-scale datasets, resulting in higher initial epistemic uncertainty compared to the ViT model. Consequently, BNDL’s capability to capture and address epistemic uncertainty is more critical for the ResNet model, leading to more pronounced improvements relative to the ViT-based model.Further, while ViT based model has already reduced epistemic uncertainty with the pre-training stage, BNDL still provides a performance boost through its probabilistic modeling approach.
> >
> > [1] Eric Wong, Shibani Santurkar, and Aleksander Madry. Leveraging sparse linear layers for debuggable deep networks. In International Conference on Machine Learning, pp. 11205–11216. PMLR, 2021.

---

### Meta-Review · Area_Chair_mrRm · 2024-12-20

**Metareview:**

This paper proposes a Bayesian neural network with a non-negative matrix factorization decision layer, called BNDL, to improve model uncertainty estimation and interpretability. While the reviewers generally agree that the paper is well-written and the methodology is sound, they raised several concerns and suggestions for improvement, including the necessity of comparing BNDL to non-Bayesian models, the validity of the disentanglement metric used, and the rigor of the theoretical analysis, questioning the comparison to non-Bayesian models and suggesting that disentanglement does not imply interpretability, and pointing out that the improvement in performance is not significant and that the PAvPU metric may be questionable, and suggesting that the theoretical analysis is not rigorous and that more metrics should be included to evaluate the performance of BNDL.

The authors responded to these concerns by arguing that comparing to non-Bayesian models is necessary to demonstrate the applicability of BNDL, providing additional visualizations and comparisons to other models to support their claims, and updating the presentation to Proposition and Analysis, and also providing more details on the experimental setup and calculation of PAvPU, and citing related works on DNNs and non-negative matrix factorization, and with some revisions to address the concerns and suggestions.

Hence, after discussion with the authors and among themselves, the reviewers find the paper much improved, with three reviewers leaning towards acceptance and only one slightly towards rejection. However, as far as the AC can tell, the latter reviewer's concerns seem to have been addressed by the authors' response. We are therefore happy to accept the paper. We would still like to encourage the authors to address all the reviewers' comments in the camera-ready version.

**Additional Comments On Reviewer Discussion:**

see above

---

### Decision · Program_Chairs · 2025-01-22

Accept (Poster)